# Dermomyotome-derived endothelial cells migrate to the dorsal aorta to support hematopoietic stem cell emergence

**Pankaj Sahai-Hernandez[1†], Claire Pouget[1†‡], Shai Eyal[1†], Ondrej Svoboda[1,2†], Jose Chacon[1], Lin Grimm[1], Tor Gjøen[3], David Traver[1]***

[1]Department of Cell and Developmental Biology, University of California, San Diego, La Jolla, United States; [2]Department of Cell Differentiation, Institute of Molecular Genetics, Academy of Sciences of the Czech Republic v.v.i, Prague, Czech Republic; [3]Department of Pharmacy, University of Oslo, Oslo, Norway

*For correspondence:
dtraver@ucsd.edu

†These authors contributed equally to this work

Present address: ‡Angiocrine Bioscience, San Diego, United States

Competing interest: The authors declare that no competing interests exist.

**Abstract** Development of the dorsal aorta is a key step in the establishment of the adult blood-forming system, since hematopoietic stem and progenitor cells (HSPCs) arise from ventral aortic endothelium in all vertebrate animals studied. Work in zebrafish has demonstrated that arterial and venous endothelial precursors arise from distinct subsets of lateral plate mesoderm. Here, we profile the transcriptome of the earliest detectable endothelial cells (ECs) during zebrafish embryogenesis to demonstrate that tissue-specific EC programs initiate much earlier than previously appreciated, by the end of gastrulation. Classic studies in the chick embryo showed that paraxial mesoderm generates a subset of somite-derived endothelial cells (SDECs) that incorporate into the dorsal aorta to replace HSPCs as they exit the aorta and enter circulation. We describe a conserved program in the zebrafish, where a rare population of endothelial precursors delaminates from the dermomyotome to incorporate exclusively into the developing dorsal aorta. Although SDECs lack hematopoietic potential, they act as a local niche to support the emergence of HSPCs from neighboring hemogenic endothelium. Thus, at least three subsets of ECs contribute to the developing dorsal aorta: vascular ECs, hemogenic ECs, and SDECs. Taken together, our findings indicate that the distinct spatial origins of endothelial precursors dictate different cellular potentials within the developing dorsal aorta.

## Editor's evaluation

In this important study, the authors identify an additional source of dorsal aorta endothelium derived from the somites that are conventionally thought of as defined blocks of skeletal muscle. The authors present convincing data that these "somite-derived endothelial cells" (SDECs) arise from bipotential precursors in the dermamyotome that give rise to endothelium or muscle. The authors conclude that these cells support but do not produce emergent hematopoietic stem cells in the dorsal aorta, findings that will be of interest to stem cell and developmental biologists.

## Introduction

The primitive vascular network, which integrates into all organ systems in the developing organism, arises from endothelial precursors termed angioblasts (*Risau and Flamme, 1995*). To form a functional vascular network, angioblasts must first differentiate into a variety of distinct arterial and venous endothelial cell (EC) types, including hemogenic, endocardial, and blood brain barrier ECs (*Aird, 2007; Herbert and Stainier, 2011*). EC differentiation is thought to initiate midway through somitogenesis,

during the migration of angioblasts to the embryonic midline, where they coalesce to form the vascular tube (*Herbert et al., 2009*; *Isogai et al., 2003*; *Jin et al., 2005*). The current view is that angioblasts are initially equipotent and undergo successive steps of differentiation that, along with cues from local microenvironments, give rise to specialized subsets of ECs (*Atkins et al., 2011*; *Marcelo et al., 2013*). This view, however, does not consider possible differences in EC function due to different developmental origins. Instead, exposure to embryonic signaling cascades mediated via Wnt (*Hübner et al., 2017*), Hedgehog (Hh) (*Gering and Patient, 2005*; *Vokes and McMahon, 2004*; *Wilkinson et al., 2012*; *Williams et al., 2010*), Vascular Endothelial Growth Factor (VEGF) (*Casie Chetty et al., 2017*; *Hong et al., 2006*; *Lawson et al., 2003*; *Lawson et al., 2002*; *Wythe et al., 2013*), and Notch molecules are thought to differentially instruct equipotent angioblasts to each distinct endothelial cell fate (*Fang et al., 2017*; *Lawson et al., 2001*; *Siekmann and Lawson, 2007*; *Zhong et al., 2001*).

Lateral plate mesoderm (LPM) is known to be the primary source of ECs across vertebrate phyla (*Potente and Mäkinen, 2017*). However, recent findings suggest that ECs can arise from distinct mesodermal derivatives, including extraembryonic-derived erythromyeloid progenitors (EMPs) that contribute extensively to the murine kidney vasculature (*Plein et al., 2018*). Furthermore, lineage tracing studies in zebrafish have demonstrated that ECs supportive of hematopoiesis can derive from endoderm (*Nakajima et al., 2023*). To better understand the development of EC subsets, we performed single-cell RNA sequencing (scRNA-seq) of ECs purified by flow cytometry over a range of time points during zebrafish embryogenesis. Following the end of gastrulation, the earliest developmental timepoint that nascent ECs can be identified, a variety of distinct molecular signatures were present, including those similar to kidney-specific ECs, brain-specific ECs, and paraxial mesoderm-derived ECs. These data suggest that specific EC fates may be specified much earlier than previously appreciated and that embryonic origins may dictate the development of tissue-specific EC subsets.

Within these datasets, we detected a signature indicative of paraxial mesoderm (PM) origins; we thus further explored this possibility since previous findings suggested that somite-derived ECs (SDECs) are intimately involved with both the development of the dorsal aorta and the emergence of hematopoietic stem and progenitor cells (HSPCs; *Nguyen et al., 2014*; *Pouget et al., 2006*). There is evidence in rodents and other amniotes that PM generates a contingent of ECs that contribute to embryonic vasculature (*Ambler et al., 2001*; *Esner et al., 2006*; *Noden, 1989*; *Pardanaud et al., 1996*; *Pouget et al., 2008*; *Pouget et al., 2006*; *Wilting et al., 1995*; *Yvernogeau et al., 2012*). Specifically, these ECs arise from a transient somitic compartment known as the dermomyotome (*Eichmann et al., 1993*; *Ema et al., 2006*; *Pouget et al., 2008*; *Yvernogeau et al., 2012*), where skeletal hypaxial muscle progenitors (skMPs) reside (*Tozer et al., 2007*). In the chick embryo, SDECs migrate to the dorsal aorta to replace hemogenic endothelium that exits the ventral aorta as emerging HSPCs (*Pouget et al., 2006*). A recent report in zebrafish has also suggested the existence of an endothelial-producing compartment (termed 'endotome') within the central somite that generates SDECs that migrate to the dorsal aorta (*Nguyen et al., 2014*). Whereas experiments in birds have suggested that SDECs only replace hemogenic endothelium, the experiments in zebrafish by *Nguyen et al., 2014* suggested that SDECs may help induce the HSPC program. However, the genetic profile of SDECs and their possible role in HSPC induction remain poorly understood.

Here, using a combination of molecular, genetic, and computational approaches, we characterize a rare population of SDECs in zebrafish that emerges from the trunk dermomyotome. Trunk SDECs migrate and contribute exclusively to rostral regions in the dorsal aorta. Within the somite, EC-fate acquisition occurs sequentially, concomitant with the epithelialization of each somite and the migration of angioblasts toward the midline of the embryo. We show that Wnt signaling is a key regulator of the distribution of ECs within the somite, whereas Notch signaling is necessary for skeletal muscle progenitor cell maintenance. Finally, epistasis experiments indicate that SDECs arise from bipotent precursors within the somite, with skMPs showing competency to become ECs in a *meox1-* and *npas4l* (*cloche Stainier et al., 1995*) – dependent manner. While SDECs integrate into the dorsal aorta, indelible lineage tracing demonstrates that they do not harbor hematopoietic potential. Instead, they appear to act at least in part as a developmental niche to facilitate HSPC emergence. Collectively, our findings indicate that distinct EC subsets are molecularly and functionally distinct as early as the end of gastrulation and describe a cellular mechanism by which the somite regulates the production of hematopoietic precursors.

## Results

### Molecular differences in endothelial cells underlie cellular diversity within the vasculature

In zebrafish, most ECs originate from the LPM (*Jin et al., 2005*). However, recent studies have suggested that somites also produce ECs that integrate into the vascular cord (*Nguyen et al., 2014*), but the role and nature of these ECs remain incompletely defined. To better understand how different ECs are specified in the zebrafish embryo, we utilized single-cell RNA sequencing (scRNA-seq) on a collection of purified ECs from distinct vascular transgenic animals to represent diverse endothelial cohorts. Specifically, we purified cells via fluorescence-activated cell sorting (FACS) from *TgBAC(et-v2:Kaede)^ci6*, *Tg(fli1:DsRed)^um13*; *Tg(tp1:GFP)^um14*, and *Tg(drl:H2B-dendra)* (hereafter termed mixed-vasculature) (*Kohli et al., 2013*; *Mosimann et al., 2015*; *Parsons et al., 2009*; *Villefranc et al., 2007*) between 22 and 24 hpf, and performed scRNA-seq on each sample. Following quality control (QC), we merged the individual datasets. Altogether, we obtained 1994 single cells that belonged to the endothelial lineage and an additional 5446 erythroid and myeloid cells, which we omitted from our analyses as our focus was on the EC subsets. Following unsupervised clustering of single-cell transcriptomes, we identified erythroid cells, myeloid cells, and eight distinct endothelial cell clusters (*Figure 1A and B*), which we named based on known marker genes within established tissue lineages (*Figure 1—source data 1* and *Figure 1—figure supplement 1*). First, we identified a sizeable endothelial cluster, general endothelium (GE), that co-expressed canonical endothelial genes and a putative hemogenic endothelium (HE) cluster that co-expressed endothelial and hematopoietic genes, including *etv2, dab2,* and *stab2*. We identified a smaller EC cluster corresponding to pre-HSCs based on expression of genes such as *cmyb, cebpa,* and *gfi1aa*. We found two clusters, brain vascular endothelial cells 1 and 2 (BVECs-I and BVECs-II), that co-expressed brain and neuronal-associated genes, including *tncb, elavl3,* and *fmoda*. Another cluster likely represents endocardial EC (EEC) based upon co-expression of canonical heart genes, including *gata5, hand2,* and *spock3*. A kidney vascular EC (KVECs) cluster co-expressed kidney-associated genes, such as *pax2a, ap1m2,* and *myh9a*. Finally, we identified an EC cluster that contained the expression of paraxial mesoderm (PM) signature genes, including *igf2a, fbn2b,* and *fn1a*. This PM signature suggested that this population may represent somite-derived endothelial cells (SDECs). We used differential expression analysis among clusters to identify distinct gene programs which are enriched within each specific subset (*Figure 1B* and *Figure 1—source data 1*).

To test our clustering predictions and verify the somitic origin of the SDEC cluster, we compared its genetic signature to annotated genes expressed within the somite compartment based on the AmiGO annotation database (*Consortium, 2019*; *Figure 1—source data 2*). Our comparative analysis resulted in 32 genes that were commonly expressed within the SDEC fraction and the AmiGO annotation database (*Figure 1C′ and C″*). Interestingly, several of these 32 genes were enriched within the SDEC cluster (*Figure 1C″* white boxed genes and 1D). We repeated our analysis on the BVEC-I and KVEC clusters. To validate their cluster identity, we compared their transcriptomes to the annotated brain and kidney genes on AmiGO. Our comparative analysis resulted in 32 and 63 common genes, respectively (*Figure 1—figure supplement 2A–C″* and *Figure 1—figure supplement 3A–C″*). Furthermore, as in the SDEC cluster, our analyses confirmed that many of these genes were enriched in both clusters (*Figure 1—figure supplement 2D* and *Figure 1—figure supplement 3D*). Thus, our scRNA-seq analysis approach identified various tissue-specific EC subsets in the 22–24 hpf embryo, including HE, pre-HSCs, and a rare population of SDECs.

### Identified EC populations can be traced back as early as the tailbud stage

After identifying the discrete genetic signatures of each endothelial cell cluster at 22–24 hpf, we queried if these signatures could also be identified at earlier stages. We followed the same protocol to purify and scRNA-seq samples collected at different developmental stages, namely *drl:H2B-dendra^+* embryos at tailbud (10 hpf) and 12 somite stages (ss) (15 hpf), as well as *etv2:Kaede^+* embryos at 15 ss (16.5 hpf) (*Figure 2A–D*). Next, we cross-referenced the transcriptomes of all EC subsets from the mixed vasculature 22–24 hpf samples (*Figures 1A and 2A*) with those of cells purified from earlier tailbud, 12-, and 15-ss samples and clustered them. We identified EC clusters with distinct

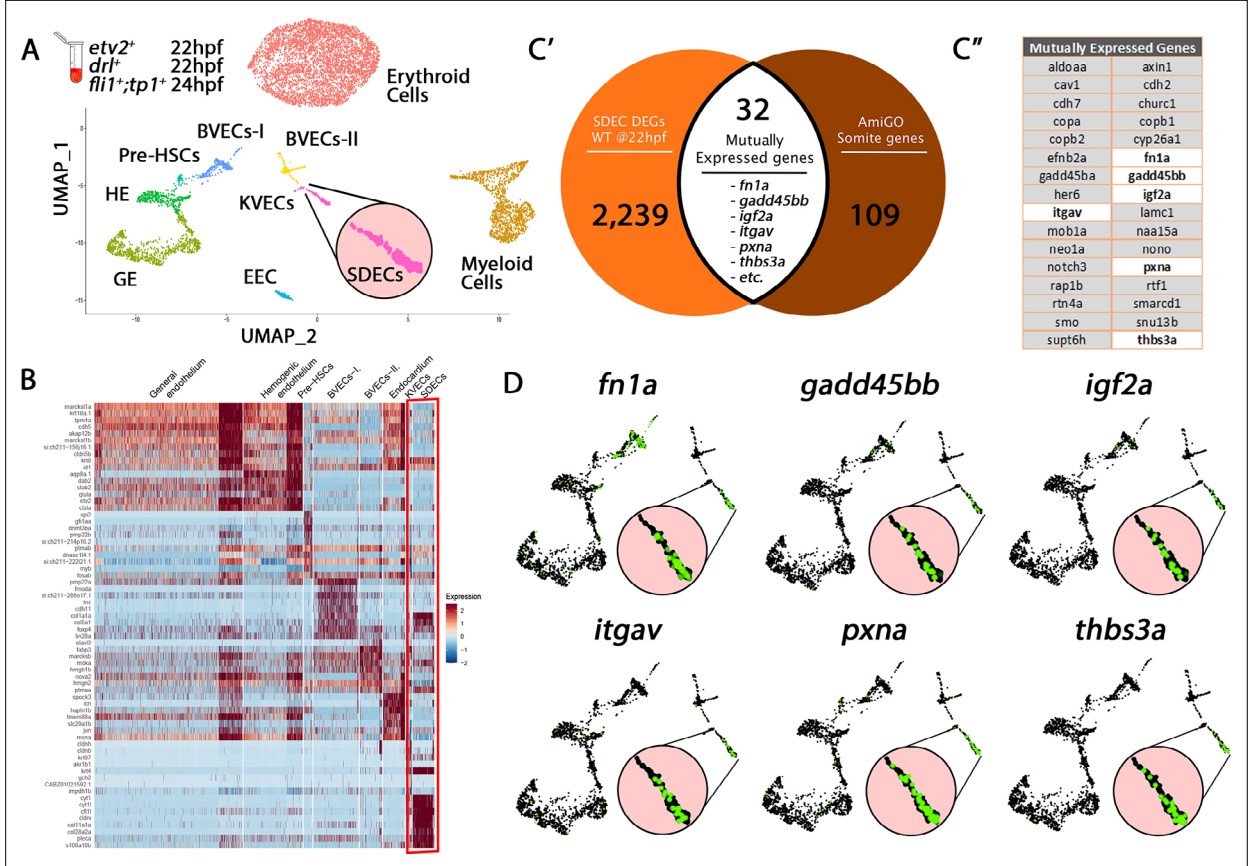

**Figure 1.** Cell-type-specific endothelial cell markers highlight cellular diversity within the vasculature. (**A**) Uniform manifold approximation projection (UMAP) plots of scRNA-seq data of total endothelial lineage cells collected from *TgBAC(etv2:Kaede)^ci6*, *Tg(fli1:DsRed)^um13*; *Tg(tp1:GFP)^um14*, and *Tg(drl:H2B-dendra)* embryos at 22–24 hpf. Clusters were named according to their gene expression: Erythroid, Lymphoid, General Endothelium (GE), Hemogenic Endothelium (HE), Pre-HSCs, Brain Vascular Endothelial Cells (BVECs-I and BVECs-II), Kidney Vascular Endothelial cells (KVECs), Endocardial Endothelial Cells (EECs), and somite-derived endothelial cells (SDECs). Color-coded marker gene expression levels are shown on corresponding clusters. A pink circle highlights the SDEC cluster. (**B**) Expression heatmap of 22–24 hpf single-cell transcriptome shows the top predicted differentially expressed marker genes across the different clusters. A red box highlights the SDEC cluster. (**C', C''**) A list of somite-annotated genes was curated from the AmiGo annotation database and compared with the SDEC transcriptome. 32 genes were commonly expressed. Interestingly, several of these 32 genes were enriched within the SDEC cluster (**C''**; white boxed genes, **D**; enlarged circles).

The online version of this article includes the following source data and figure supplement(s) for figure 1:

**Source data 1.** Transcriptomes of all endothelial cell clusters, myeloid, and erythroid cells.

**Source data 2.** Comparison between Genes expressed in EC clusters (e.g. SDECs) and gene annotation of the same anatomical structure (e.g., somite) based on annotation from AmiGO (*Consortium, 2019*).

**Figure supplement 1.** Cluster identity was assigned based on known marker genes.

**Figure supplement 2.** Comparison of BVECs-I cluster genes to brain annotated genes validates cluster origin.

**Figure supplement 3.** Comparison of KVEC Cluster genes to kidney annotated genes validates cluster origin.

transcriptomes as early as the tailbud stage, suggesting that EC specification started by the end of gastrulation and the initiation of somitogenesis. While the EC clusters were still heterogeneous prior to the 12 ss stage, we could identify most EC subsets, including HE, pre-HSC, KVECs, BVECs, and SDECs, after the 15 ss when their transcriptomes became more defined (*Figure 2B*). Together, these data suggest a gradual EC specification process that begins as early as the tailbud stage (*Figure 2B–D*).

To explore how early tissue-specific EC segregation occurs, we compared the changes in expression patterns of EC populations from early *TgBAC(etv2:Kaede)^ci6* 15 ss and later 22 hpf scRNA-seq data. Specifically, we focused on the changes in expression in the 32 overlapping genes between the scRNA-seq transcriptome data and the AmiGo annotated genes (*Figure 1C'* and *Figure 1—source data 2*). From the 32 SDEC genes compared, 16 genes were upregulated in the *etv2:Kaede^+* 22 hpf

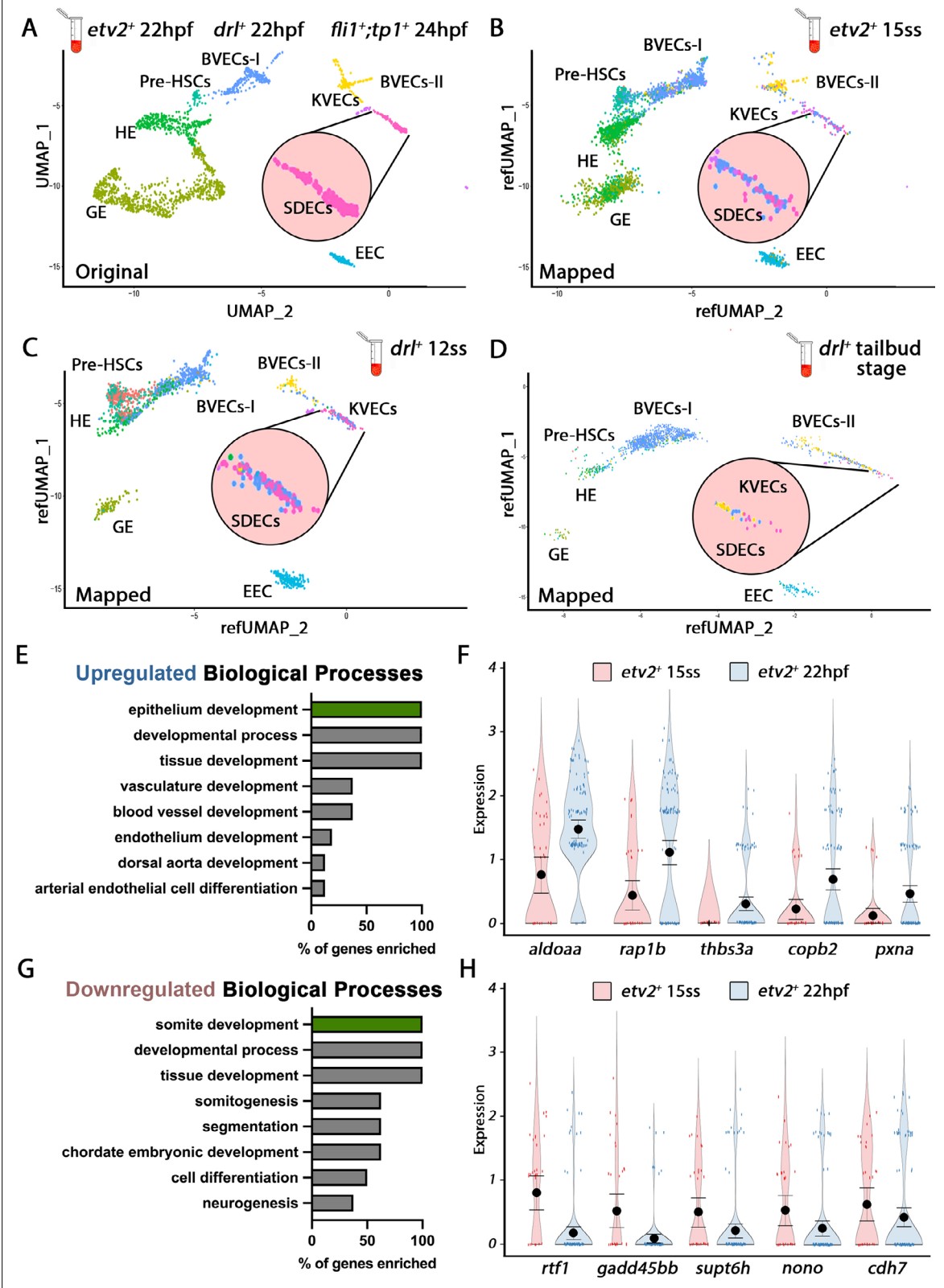

**Figure 2.** Cellular diversity within the vasculature can be traced back to the tailbud stage. (**A**) Uniform manifold approximation projection (UMAP) plots of scRNA-seq data of total endothelial lineage cells collected from *TgBAC(etv2:Kaede)^ci6^, Tg(fli1:DsRed)^um13^; Tg(tp1:GFP)^um14^*, and *Tg(drl:H2B-dendra)* embryos at 22–24 hpf. Clusters were named according to their gene expression: General Endothelium (GE), Hemogenic Endothelium (HE), Pre-HSCs, Brain Vascular Endothelial Cells (BVECs-I and BVECs-II), Kidney Vascular Endothelial cells (KVECs), Endocardial Endothelial Cells (EEC), and somite-

*Figure 2 continued on next page*

*Figure 2 continued*

derived endothelial cells (SDECs). Color-coded marker gene expression levels are shown on corresponding clusters. A pink circle highlights the SDEC cluster. (**B–D**) Referenced uniform manifold approximation projection (RefUMAP) plots of scRNA-seq data of total endothelial lineage cells collected from *etv2:Kaede*+ embryos at 15 ss (**B**) and *drl:H2B-dendra*+ embryos at 12 ss (**C**) and tailbud stage (**D**). By cross-referencing the transcriptomes of EC subsets at each developmental stage to the 22–24 hpf ECs, we identified EC clusters with distinct transcriptomes as early as the tailbud stage. (**E–H**) Comparison of expression patterns of EC populations from early *TgBAC(etv2:Kaede)*ci6 15 ss, and later 22 hpf *etv2:Kaede*+ in the 32 overlapping genes between the SDEC transcriptome data and the AmiGo somite annotated genes. (**E,F**) Representative genes that were upregulated in the *etv2:Kaede*+ 22 hpf samples compared to the 15 ss sample (**F**) and their suggested role in EC differentiation, according to GO biological processes (**E**). (**G,H**) Representative genes that were downregulated in the *etv2:Kaede*+ 22 hpf samples compared to the 15 ss sample (**H**) and their suggested role in somitogenesis, according to GO biological processes (**G**). The expression and downregulation of somitic genes within *etv2*+ ECs between 15 ss and 22 hpf highlight their somitic origin and loss of myogenic cell fate.

The online version of this article includes the following figure supplement(s) for figure 2:

**Figure supplement 1.** Differentially expressed genes between early and late ECs in BVECs-I or KVECs clusters highlight an early commitment to EC fate.

sample compared to the 15 ss sample (*Figure 2E and F*), whereas 16 genes were downregulated (*Figure 2G and H*). GO function analysis indicated a role for upregulated genes in arterial EC differentiation, whereas downregulated genes were involved in somitogenesis (*Figure 2E and G*, respectively). The expression and downregulation of somitic genes within *etv2*+ ECs between 15 ss and 22 hpf highlight their somitic origin and the loss of myogenic cell fate of these cells as they emerge from the somite compartment.

We repeated our analysis on the BVEC-I and KVEC clusters to validate this observation by comparing samples from *etv2:Kaede*+ 15 ss and 22 hpf. In the BVEC-I cluster, we found 17 upregulated genes that indicate epithelium development, whereas seven genes indicating brain development were downregulated (*Figure 2—figure supplement 1A–D*). In the KVEC cluster, we found 18 upregulated genes indicative of epithelial cell differentiation in the kidney and 14 downregulated genes indicating kidney development (*Figure 2—figure supplement 1E–H*). As with SDECs, both BVEC-I and KVEC cells showed evidence of their commitment to a brain and kidney EC fate as early as 15 ss.

## Rare endothelial cells emerge from trunk somites located above the dorsal aorta

Our transcriptome analysis identified that SDECs arise as a distinct EC population as early as the tailbud stage. Next, we aimed to characterize the development of SDECs and their potential to contribute to DA formation at later stages. To label somitic cells and follow their trajectories, we crossed a photoconvertible *Tg(actb2:nls-Eos)* animal with a *Tg(fli1:eGFP)*y1 vasculature reporter line (*Cruz et al., 2015*; *Isogai et al., 2003*). Resulting embryos ubiquitously express a photoconvertible nuclear GFP that can be instantly converted to RFP by UV exposure. To determine which somites generate SDECs and to quantify how many SDECs are made by each somite pair, we collected embryos at different somitogenesis stages ranging from 4 ss to 18 ss and converted single somite pairs using directed UV exposure via confocal microscopy. We avoided converting any LPM-derived EC on the lateral or ventral sides of the somite (*Figure 3A and A'*). Embryos were allowed to develop to 32–36 hpf, then imaged and examined for the presence of RFP+ SDECs within the dorsal aorta (*Figure 3B*). We observed that the trunk somites (numbers 5–18), located above the yolk tube extension generated the most SDECs. In contrast, photoconverted somites anterior to the 4th somite rarely gave rise to SDECs (*Figure 3C and D*). In sum, most SDECs were generated by somites 5–18. Each somite pair contributed between 0–6 SDECs to the DA (*Figure 3D* and *Figure 3—source data 1*). Interestingly, we occasionally noticed SDECs within the roof of the DA, suggesting a migratory path that passes through the roof of the DA before moving to the floor of the DA, consistent with previous reports (*Zhao et al., 2022*). On rare occasions, we also detected SDECs in the posterior caudal vein or intersomitic vessels, consistent with previous findings (*Nguyen et al., 2014*).

## Somites give rise to *etv2*+ endothelial cells concomitant with somite epithelialization

Next, we aimed to further characterize the development of SDECs by following *etv2:eGFP*+ cells and their trajectories from the somite. As the SDEC subset became more distinct between 12–15 ss,

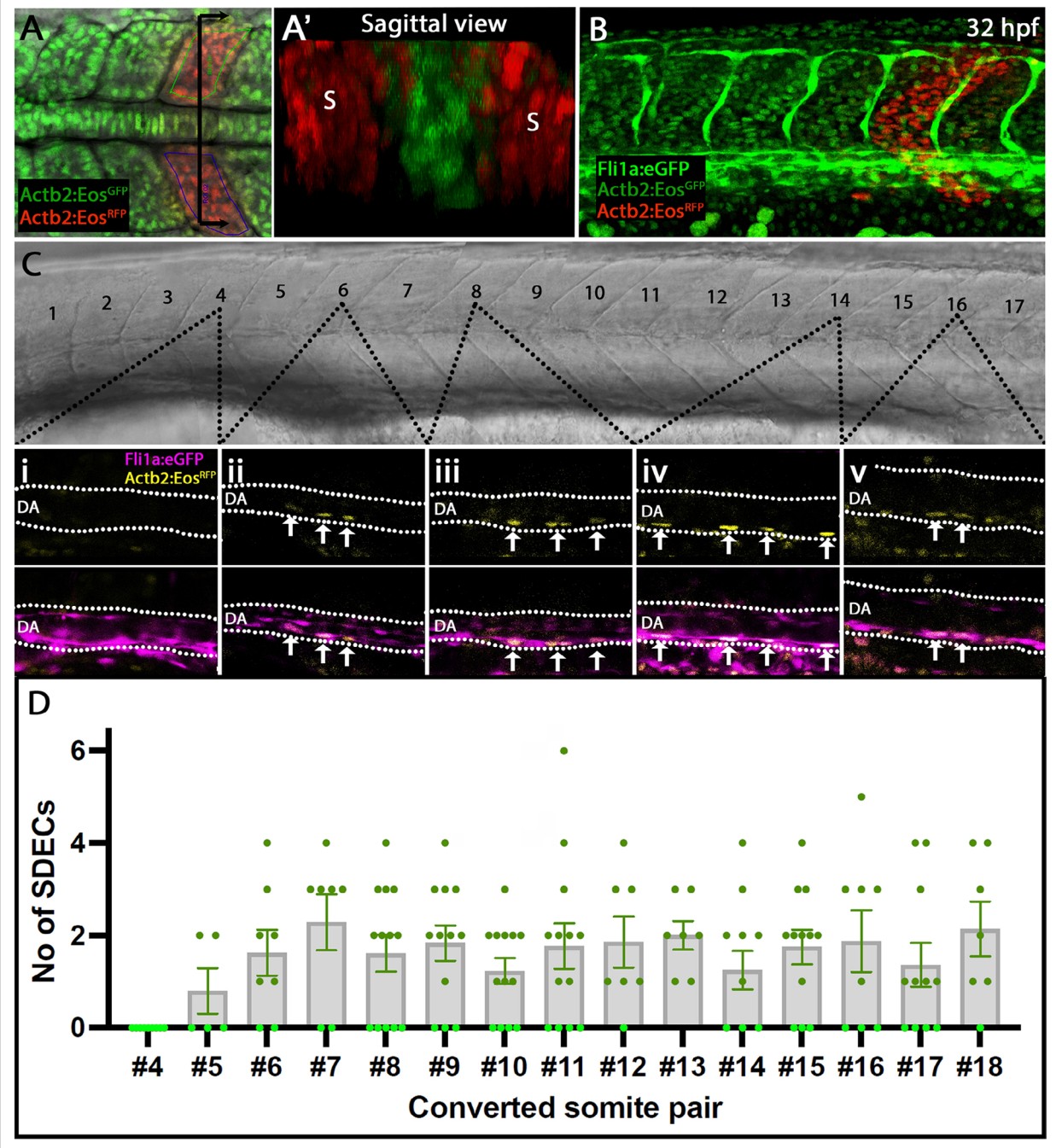

**Figure 3.** Rare SDECs emerge from trunk somites and migrate to the dorsal aorta. (**A–D**) *Tg(actb2:nls-Eos)*; *Tg(fli1:eGFP)^y1* embryos were collected at developmental stages ranging from 4 to 18 ss. (**A**) Newly developed posterior somite pairs were selected by setting a region of interest and photoconverted by UV light. (**A'**) A sagittal section through a pair of converted somite showing somite-specific conversion and lack of converted LPM-derived ECs. (**B**) At 32 hpf, embryos were laterally staged, and images of the dorsal aorta taken. SDECS were quantified in the dorsal aorta by examining individual z-stacks and visualizing colocalization of *fli1+*; *actb2:nlsEos^RFP* converted cells in *Tg(actb2:nls-Eos)*; *Tg(fli1:eGFP)^y1* embryos (**Ci-Cv**). In *fli1:eGFP^-* embryos, we identified SDECs by observing *actb2:nlsEos^RFP* cells on the floor of the DA based on the brightfield channel. (**D**) We observed that trunk somites (numbers 5–18), located above the yolk tube extension, generated the most SDECs. Each somite pair contributed between 0–6 SDECs to the DA. s, somites; DA, dorsal aorta. In each converted somite pair, $n \geq 6$, with each point representing the SDEC count from one embryo. The median for each somite pair is indicated as a column, and the standard error of the mean (SEM) is indicated as an error bar.

The online version of this article includes the following source data for figure 3:

**Source data 1.** A table summarizing all converted somite pairs and SDECs found in *Tg(actb2:nls-Eos)*; *Tg(fli1:eGFP)^y1* embryos that were included in the final SDECs quantification assay.

we focused our analysis on that developmental time frame. To do so, we used confocal time-lapse imaging of *Tg(etv2.1:eGFP)*<sup>zf372</sup> embryos (*Veldman and Lin, 2012*) injected with *mOrange2:CAAX* mRNA. These embryos have early ECs marked by GFP with cell boundaries demarcated by mOrange2. As previously described (*Veldman and Lin, 2012*), beginning at the 10 ss, we observed a line of *etv2:eGFP*<sup>+</sup> along the most medial part of the LPM (*Figure 4A*). We did not detect any *etv2:eGFP*<sup>+</sup> in the somites at this stage. Initiating at the 12 ss, we noted *etv2:eGFP*<sup>+</sup> in the lateral lip of the somitic compartment (*Figure 4B*). After the onset of *etv2:eGFP* expression, within the next 3 hr, somite-derived *etv2:eGFP*<sup>+</sup> rounded up and delaminated from the somite, then integrated into the cohort of LPM-derived *etv2:eGFP*<sup>+</sup> that migrated toward the midline (*Figure 4C–E*). Time-lapse imaging of *Tg(etv2.1:eGFP)*<sup>zf372</sup>; *Tg(phlbd1:Gal4-mCherry)* embryos resulted in similar observations that support the emergence of ECs from the somites as early as 12 ss (*Figure 4—video 1* and *Figure 4—video 2*).

In mice and chick, SDECs emerge from the same region as skeletal muscle progenitor cells in the hypaxial dermomyotome compartment (*Pouget et al., 2006*; *Tozer et al., 2007*). To examine the spatial origins of SDECs in the zebrafish, we performed double fluorescent in situ hybridization (FISH) for the endothelial marker *etv2* and the skeletal muscle progenitor marker *pax3a* (*Relaix et al., 2005*). At 12 ss, we observed colocalization of *etv2* and *pax3a* within rare cells in the somite (*Figure 4F*). However, FISH for *etv2* and *myod*, a marker of differentiated muscle cell types (reviewed in *Hernández-Hernández et al., 2017*), did not show colocalization (*Figure 4G*). These results suggest that SDECs emerge within the somite from precursor cells shared with the muscle lineage. We observed that somitic *etv2*<sup>+</sup> were localized specifically within the dermomyotome region, the muscle progenitor cell compartment of the somite. As this was previously shown in other organisms, we suggest that a conserved mechanism of SDEC generation is shared amongst vertebrates (*Mayeuf-Louchart et al., 2016*; *Mayeuf-Louchart et al., 2014*; *Pouget et al., 2006*).

## The dermomyotome contains progenitors with muscle and endothelial potential

Since our results above suggested the presence of bipotent progenitors with competence for muscle and endothelial cell differentiation, we sought to determine if blocking skeletal muscle differentiation may lead to enhanced SDEC generation by knocking down *mesenchyme homeobox 1* (*meox1*), an essential regulator of muscle cell formation (*Mankoo et al., 2003*). To knock down *meox1*, we used a *meox1* translation blocking morpholino (MO) that can block translation of a functioning Meox1 protein but does not affect the transcription of a viable *meox1* mRNA; thus, it can still be detected by an RNA probe. Using FISH to label *meox1* and *etv2* expression, we observed ectopic expression of *etv2* in morphant animals and an increased number of SDECs (*Figure 5A–F*). Like normal SDECs, ectopic *etv2:eGFP*<sup>+</sup> emerged at 12 ss. However, in contrast to wild type, *meox1* morphants showed SDEC generation as late as 23 hpf in *Tg(phldb1:mCherry)*; *Tg(etv2.1:eGFP)*<sup>zf372</sup> embryos (*Distel et al., 2009*; *Kobayashi et al., 2014*; *Figure 5G and H*). Thus, *meox1* loss of function leads to enhanced and prolonged production of SDECs.

Previous work in mice identified the Notch signaling pathway as a positive regulator of muscle fate in the somite (*Mayeuf-Louchart et al., 2014*). We tested the requirement for Notch signaling in zebrafish SDEC generation by knocking down the essential *notch* regulator *mindbomb (mib)* (*Itoh et al., 2003*) in *Tg(etv2.1:eGFP)*<sup>zf372</sup> animals and examining SDEC formation between 12–14 ss. While loss of *meox1* alone led to the ectopic formation of a few SDECs (*Figure 5I and J*), knockdown of both *mib* and *meox1* led to a profound expansion of *etv2:eGFP*<sup>+</sup> within the somite (*Figure 5K*).

To examine the role of Notch signaling more precisely in these cells, we examined *notch3*<sup>fh332/fh332</sup> animals (referred to as *notch3*<sup>-/-</sup>) since *notch3* is the primary *notch* receptor in the somites at this stage (*Kim et al., 2014*). We injected *meox1* MO into *notch3*<sup>-/-</sup> embryos and performed FISH for *meox1* and *etv2* (*Figure 5L–N*). Using this strategy, we observed that in *notch3*<sup>-/-</sup> homozygous mutant embryos, the formation of SDECs was not impaired (*Figure 5L and M*). However, following the combined loss of *notch3* and *meox1*, we observed the ectopic formation of *etv2*<sup>+</sup>; *meox1*<sup>+</sup> double positive cells by FISH (*Figure 5N* and *Figure 5—figure supplement 1*).

To determine which cell populations were explicitly affected by the loss of *notch3*, we compared the expression levels of markers for muscle progenitors (*pax3a, pax7b*), differentiated muscle cells (*myod, myog*), and differentiated endothelial cells (*etv2, fli1a*) by qRT-PCR in *notch3* mutant and heterozygous embryos at 24 hpf (*Figure 5O*). In *notch3*<sup>-/-</sup> animals, we observed a decrease in the expression of

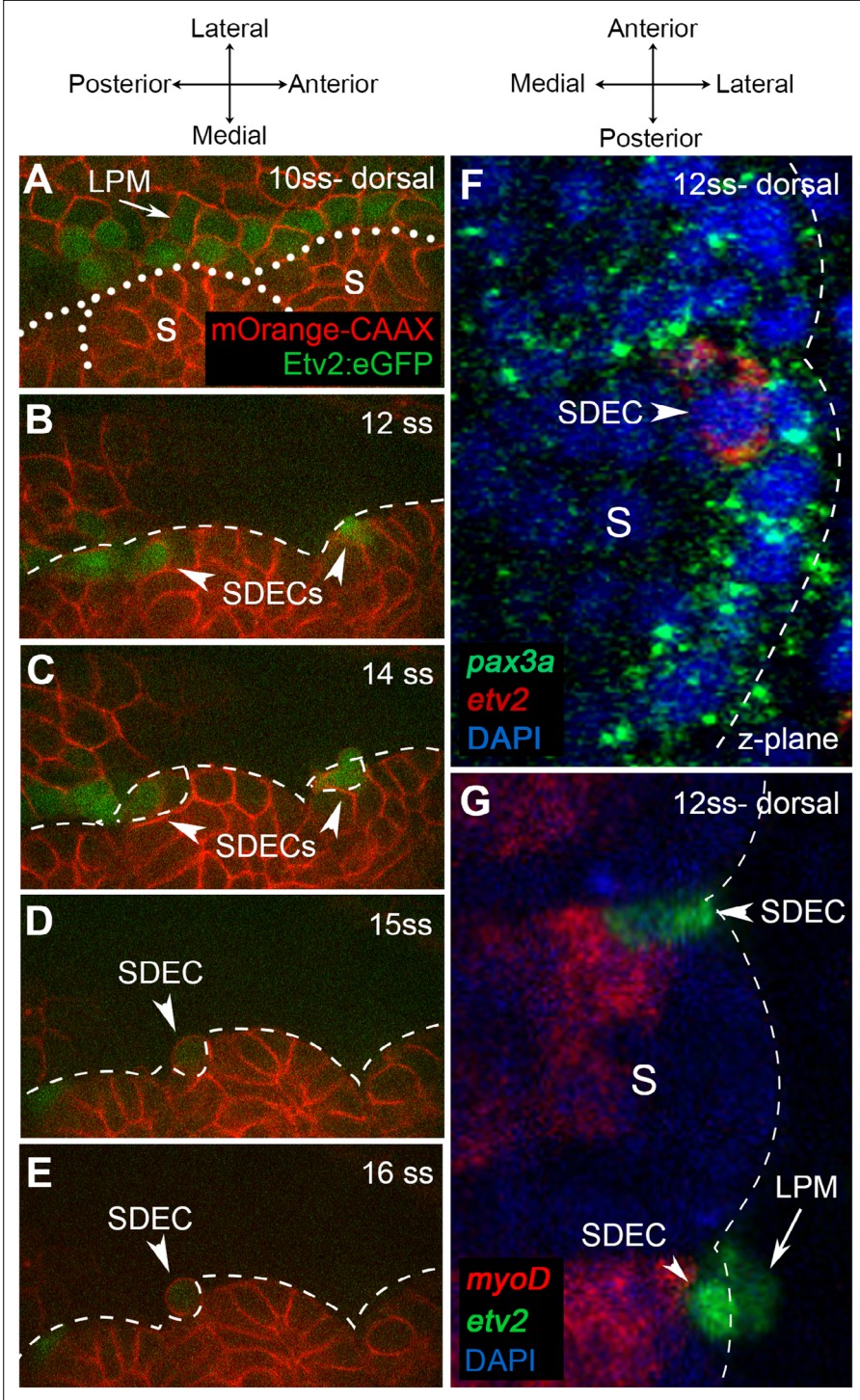

**Figure 4.** Endothelial cells emerge from the dermomyotome at 12 ss. (**A–E**) Time-lapse imaging from a dorsal view of *Tg(etv2.1:EGFP)^{zf372}* embryos injected with *mOrange:CAAX* mRNA and imaged between 10 ss and 15 ss. (**A**) The expression of Etv2:GFP⁺ cells is visible along the LPM region (arrow) at 10 ss. At this stage, no Etv2:GFP⁺ cells are visible in the somites. (**B**) Starting at 12 ss, the first Etv2:GFP⁺ SDECs are detected in the lateral lip of the dermomyotome (arrowheads). Simultaneously, the LPM Etv2:GFP⁺ cells start migrating to the midline. (**C**) Soon after emergence, SDECs change shape and become rounder (arrowheads). (**D–E**) Etv2:GFP⁺ SDECs bud off from the somite as individual cells (arrowhead). (**F**) Dorsal view of a 12 ss embryo that was submitted to double fluorescent in situ hybridization for muscle progenitor maker *pax3a* (green) and endothelial marker *etv2* (red). *pax3a* expression reveals the dermomyotome compartment that contains muscle progenitor cells. An *etv2*⁺ SDEC

*Figure 4 continued on next page*

*Figure 4 continued*

(red and arrowhead) is found in the dermomyotome, co-expressing *pax3a* (green), showing colocalization of an endothelial and muscle progenitor cell marker. We observed 1–2 *etv2*-positive cells per somite in each of the embryos examined (n=6). (**G**) Somitic *etv2*⁺ SDECs (green) do not co-express the muscle differentiation marker *myoD* (red), suggesting that *etv2* expression is restricted to the muscle progenitor region of the somite. Dashed white lines delimit somite from the LPM (arrow). We observed 1–2 *etv2*-positive cells per somite in each of the embryos examined (n=6). s, somites; LPM, lateral plate mesoderm; SDECs, somite-derived endothelial cells.

The online version of this article includes the following video(s) for figure 4:

**Figure 4—video 1.** Time-lapse imaging of *Tg(etv2.1:eGFP)ᶻᶠ³⁷²; Tg(phldb1:mCherry)* embryo between 12 and 16 ss. https://elifesciences.org/articles/58300/figures#fig4video1

**Figure 4—video 2.** Time-lapse imaging of *Tg(etv2.1:eGFP)ᶻᶠ³⁷²; Tg(phldb1:mCherry)* embryo between 12 and 16 ss. https://elifesciences.org/articles/58300/figures#fig4video2

---

muscle progenitor markers (*pax3a*, *pax7b*) concomitant with an increase in the expression of muscle differentiation markers (*myod*, *myog*) and endothelial differentiation markers (*etv2*, *fli1a*) (**Figure 5O**). Furthermore, we observed premature expression of the muscle differentiation marker *myoHII* (**Beier et al., 2011**; **Salucci et al., 2015**; **Sjöblom et al., 2008**) by antibody staining at 48 hpf in *notch3⁻/⁻* animals (**Figure 5P and Q**). Together, these results suggest that Notch signaling is dispensable for the specification of SDECs but is required for *pax3⁺* and *pax7⁺* skeletal muscle progenitor maintenance, as previously described (**Relaix et al., 2005**; **Zhang et al., 2021**). Moreover, the absence of *notch3* leads to premature differentiation of muscle and SDEC fates (**Figure 5R**).

*npas4l* is required for the formation of SDECs *npas4l (cloche)* is regarded as the most upstream gene required for blood and endothelial cell specification (**Reischauer et al., 2016**; **Stainier et al., 1995**). Therefore, we sought to determine if *npas4l* is also required for SDEC development. WISH in 12 ss embryos showed a complete absence of *etv2* expression along the embryonic A-P axis in *Cloˢ⁹/ˢ⁹* mutants (referred to as *cloche*) compared to controls (**Figure 6A–B**), showing that *npas4l* function is required for both PLM-derived ECs and PM-derived SDECs. Knockdown of *meox1* to increase SDEC differentiation from bipotent somitic progenitors was insufficient to rescue SDEC generation, indicating *npas4l* function is necessary to generate the SDEC lineage. In addition, suppressing muscle specification in the absence of *npas4l* was insufficient to induce SDEC fate (**Figure 6C–D**). Similarly, in *Tg(etv2.1:eGFP)ᶻᶠ³⁷²; cloche* embryos injected with *mib* MO and *meox1* MO, we observed a complete absence of *etv2:eGFP⁺*, compared to *cloche* control siblings (**Figure 6—figure supplement 1**).

Lastly, we performed qRT-PCR for endothelial and muscle cell genes from 48 hpf *cloche* mutant and control embryos (**Figure 6E**). As expected, *cloche* mutant embryos showed decreased endothelial gene expression (*etv2* and *fli1a*). Interestingly, we also observed a concomitant increase in the expression of skeletal muscle differentiation genes (*myod* and *myogenin*) (**Figure 6E**). Together, these results confirm that *npas4l* is required for specification of all EC subsets, including SDECs, and suggest a negative regulatory role in muscle differentiation in the shared bipotent progenitors of SDECs and skeletal muscle cells (**Figure 6F and G**).

## Wnt signaling regionalizes the formation of SDECs

Previous work has shown that Wnt signaling is required for the differentiation of muscle progenitors through activation of the required skeletal muscle factor *myf5* (reviewed in **von Maltzahn et al., 2012**). In addition, inhibition of Wnt signaling in early presomitic mesoderm leads to an increase in endothelial cells that can integrate into the zebrafish vasculature (**Veldman et al., 2013**), suggesting that Wnt signaling may also act later in the somite to balance muscle and endothelial cell production from shared muscle progenitor cells. To investigate whether Wnt signaling is active in *meox1⁺* muscle progenitor cells during SDEC development, we performed FISH for *meox1* in the background of *Tg(7xTCF-Xla.Siam:GFP)ⁱᵃ⁴*, a destabilized Wnt/TCF reporter line +. At the 12 ss, we observed cells positive for GFP and *meox1* (**Figure 7A**), indicating that Wnt signaling is active while SDEC fate decisions occur. Next, to determine if Wnt inhibition affects SDEC development, we treated *Tg(etv2.1:eGFP)ᶻᶠ³⁷²* embryos with IWP-L6, a potent inhibitor of Wnt protein secretion (**Wang et al., 2013**), from 2 to 12 ss. Inhibition of the Wnt signal was confirmed by qRT-PCR for the canonical target gene *axin2* (**Figure 7B**), which led to a decrease in *axin2* and an increase in *etv2* transcripts. In addition, we observed a slight reduction in *meox1* expression; however, it was not statistically

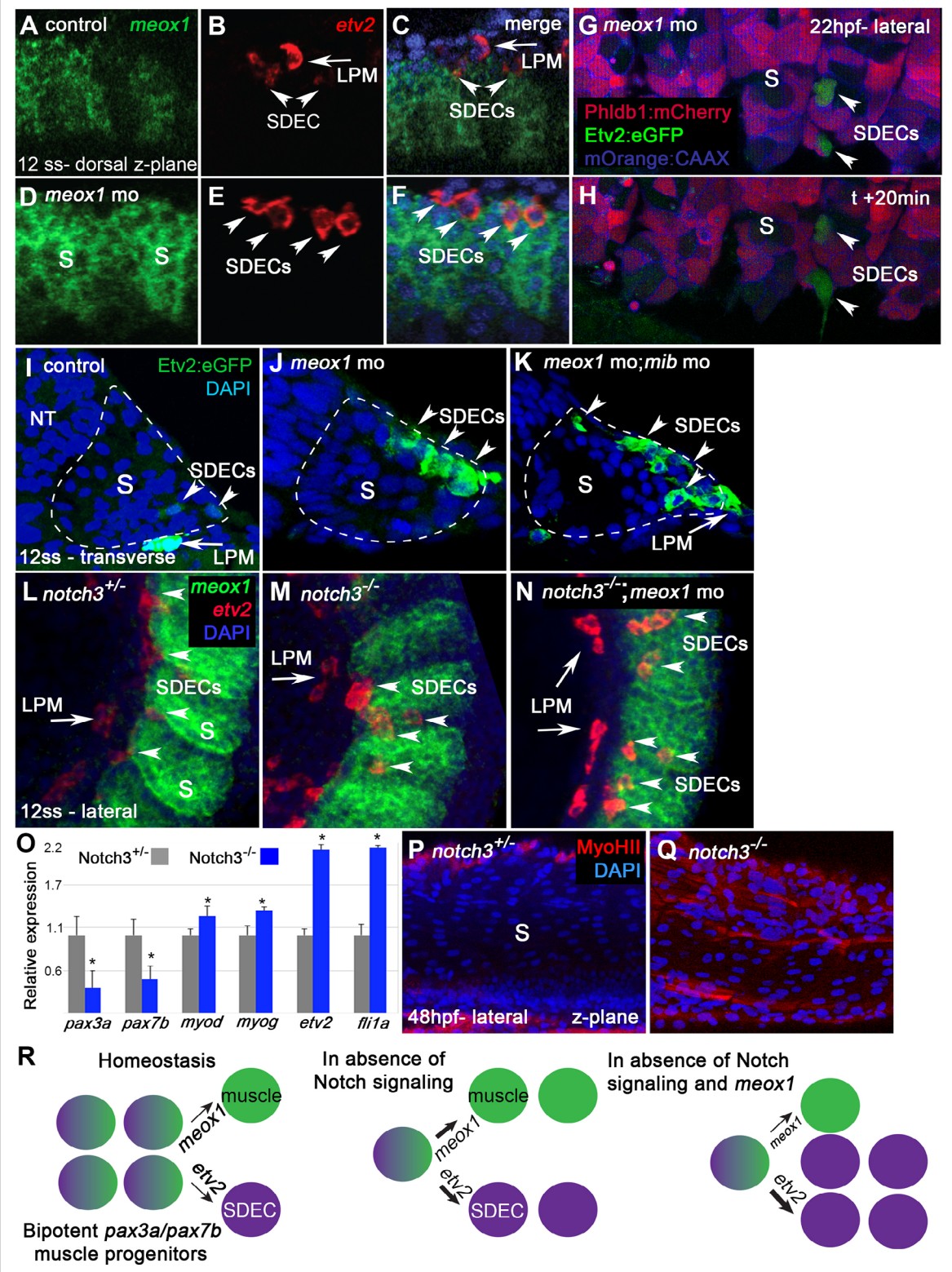

**Figure 5.** *notch* is required for the maintenance of a bipotent skeletal muscle progenitor population in the somite. (**A–F**) Dorsal view of 12 ss control (**A–C**) and *meox1* morphant embryos (**D–F**). Embryos were submitted to double fluorescent in situ hybridization for *meox1* (green) and *etv2* (red). In control and morphant embryos, *meox1; etv2* double-positive cells are detected within the somite compartment (arrowheads). (**C,F**) Overlay of *meox1* (green), *etv2* (red), and DAPI (blue). (**D–F**) Knockdown of *meox1* results in ectopic formation of double-positive cells within the somite (arrowheads). We observed 3–4 *etv2* positive cells per somite in the *meox1* morphants compared to 1–2 *etv2*-positive cells per somite in the siblings (n=3). (**G–H**) Time-

*Figure 5 continued on next page*

*Figure 5 continued*

lapse imaging of a 22 hpf *Tg(etv2.1:eGFP)^{zf372}; Tg(phldb1:mCherry)* embryo, injected with *meox1* morpholino and *mOrange2:CAAX* mRNA to delineate cell boundaries. Knockdown of *meox1* results in an extension of the period that the dermomyotome can generate Etv2:GFP^+ cells (arrowheads). (**I–K**) Cross section of 12 ss *Tg(etv2.1:eGFP)^{zf372}* embryo. In absence of *meox1* (**J**), ectopic Etv2:GFP^+ cells are visible in epithelialized layer of the somites, compared to controls (**I**). In embryos coinjected with *mib* and *meox1* morpholinos, the number of Etv2:GFP^+ cells within the somite compartment (dotted line) is substantially increased (arrowheads) (**K**), suggesting that Notch signaling is dispensable for SDEC specification. (**L–N**) Lateral view of 12 ss embryos analyzed by FISH for *meox1* (green), *etv2* (red), and DAPI (blue). In *notch3^{+/-}* heterozygote controls (**L**) and *notch3^{-/-}* mutant embryos (**M**), *etv2^+* SDECs are detected in the somites. (**N**) *notch3^{-/-}* mutant embryos co-injected with *meox1* morpholino results in ectopic formation of *etv2; meox1* double positive cells (arrowheads). We observed 2–4 *etv2*-positive cells per somite in the *notch3* mutants and >6 *etv2*-positive cells in the *notch3* mutants; Mib morphants (n=3). (**O**) qRT-PCR in 24 hpf *notch3^{-/-}* mutant embryos and sibling controls. Genetic ablation of *notch3* results in decreased expression of muscle progenitor markers *pax3a* and *pax7b*; increased expression of muscle differentiation genes, *myod* and *myog*, and endothelial markers, *etv2*, and *fli1*. Asterisks denote a statistically significant difference (p<0.05, unpaired, two-tailed Student's t-test; n=3.) (**P,Q**) *notch3^{-/-}* mutant embryos show premature expression of MyoHII in 48 hpf embryos (**Q**) compared to sibling controls (**P**). (**R**) Summary cartoon for the role of Notch signaling in the maintenance of bipotent-muscle progenitors (bipotent muscle progenitors in purple and green; muscle cells in green; SDECs in purple). s, somites; LPM, lateral plate mesoderm; SDECs, somite-derived endothelial cells.

The online version of this article includes the following figure supplement(s) for figure 5:

**Figure supplement 1.** Bipotent muscle progenitor cells contain endothelial potential that can reach the dermomyotome compartment.

significant. By examining serial sections, Wnt inhibition led to the formation of ectopic *etv2:eGFP^+* in the somites (***Figure 7C–D'***). Finally, we also determined that Wnt signaling is required for *meox1* expression (***Figure 7E–F'***). Together, these results demonstrate that Wnt signaling compartmentalizes the formation of SDECs within the somite to the most lateral region. In addition, they show that Wnt inhibition results in downregulation of *meox1* and the expansion of *etv2:eGFP^+*.

## SDECs integrate into the dorsal aorta but do not generate HSPCs

Our results confirmed previous studies in the chick (***Pouget et al., 2006***) and zebrafish (***Nguyen et al., 2014***) that ECs emerge from the somites and incorporate into the DA (***Figure 3***). Next, we were interested in understanding the role of SDECs in the developing zebrafish vasculature and hematopoiesis. To this end, we obtained a PM-specific Gal4 transgenic line, *Tg(tbx6:Gal4FF:GFP-nls)* (***Yabe et al., 2016***), which has been shown to recapitulate *tbx6* mRNA expression (referred to as *tbx6:Gal4*). We crossed this line to *Tg(UAS-Cre)* (***Butko et al., 2015***) to drive Cre recombinase only within the PM. We then crossed it to a ubiquitously expressed reporter line *Tg(actb2:loxP-BFP-loxP-DsRed)^{sd27}* (***Kobayashi et al., 2014***), which upon genetic recombination, switches from a BFP to DsRed cassette (abbreviated as A2BD for clarity). Upon confocal imaging of the triple transgenic *tbx6:Gal4; Tg(UAS-Cre); A2BD* animals at 48 hpf, we observed DsRed^+ cells within the region of the axial vasculature (***Figure 8A–C***). To confirm that these cells were SDECs arising from a *tbx6^+* somitic population, we crossed the *tbx6:Gal4; Tg(UAS-Cre)* line with a previously published endothelial-specific CFP-to-YFP switch line, *TgBAC(kdrl:LOXP-AmCyan-LOXP-ZsYellow)* (referred to as *kdrl:CSY*) (***Zhou et al., 2011***). Imaging the DA in *tbx6:Gal4; Tg(UAS-Cre); kdrl:CSY* triple transgenic embryos at 4 dpf showed YFP^+ cells corresponding to SDECs to localize preferentially to the trunk region of the dorsal aorta (***Figure 8D–G and K***). We found this labeling pattern consistent among different embryos over a range of developmental time points.

To complement these results, we performed lineage tracing using an LPM-specific switch line, *Tg(drl:Cre^{ERT2})* (***Henninger et al., 2017***). Likewise, we crossed it to *kdrl:CSY* for EC-specific tracing. We incubated *Tg(drl:Cre^{ERT2}); kdrl:CSY* embryos with tamoxifen (4-OHT) to induce Cre-based recombination between 8 and 24 hpf. At 4 dpf, confocal imaging showed complementary results to our PM-specific *tbx6^+* lineage tracing experiments. We observed that *drl*-derived YFP^+ ECs contributed to all regions of the vasculature, as would be expected from an LPM source (***Figure 8H–J and K***). However, we observed that the contribution of these cells was more robust within the posterior region of the DA and inverse to the contribution of the PM-specific, *tbx6*-labeled SDECs, which contributed ECs mainly to the trunk region of the vasculature (***Figure 8D–K***). Together, these results further demonstrate that SDECs integrate into the DA in zebrafish and appear to contribute preferentially to the trunk portion of the dorsal aorta.

Since HSPCs derive from hemogenic endothelium, specifically within the DA, we examined whether or not SDECs can generate HSPCs. We lineage traced PM-derived cells into the kidney, the adult

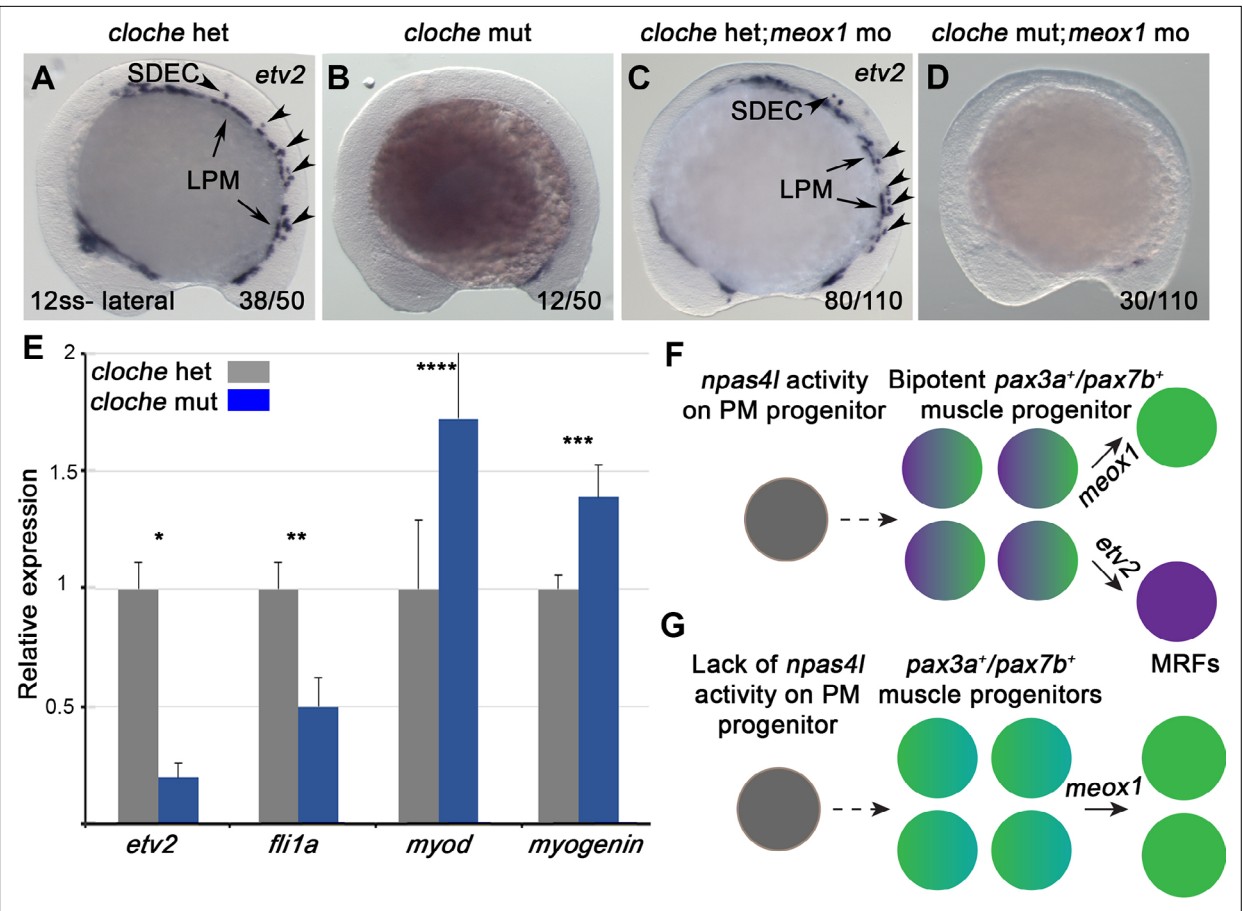

**Figure 6.** *npas4l* is required for the specification of SDECs. (**A–D**) WISH for *etv2* in 12 ss *npas4l⁻/⁻* (*cloche*) mutant and control embryos. (**B**) *cloche* mutant embryos show an absence of *etv2* expression along the A-P axis of the embryo, compared to sibling control (**A**). (**D**) Similarly, *cloche* mutant embryos injected with *meox1* morpholino show loss of *etv2* expression, compared to sibling control (**C**). (**E**) qRT-PCR of *cloche* mutant embryos shows expected loss of endothelial genes (*fli1* and *etv2*) and concomitant increase of muscle differentiation genes (*myod* and *myog*), compared to sibling control. All genes analyzed between *cloche* mutant and *cloche* het embryos showed a statistically significant difference (p<0.001, unpaired, two-tailed Student's t-test; n=3.) (**F, G**) Summary cartoon for the effect of *npas4l* on endothelial cell competence in PM progenitors (early mesoderm progenitor in grey; bipotent muscle progenitor in purple and green; muscle cells in green; endothelial cells in purple). LPM, lateral plate mesoderm; SDECs, somite-derived endothelial cells.

The online version of this article includes the following figure supplement(s) for figure 6:

**Figure supplement 1.** *npas4l* is required for the specification of SDECs.

hematopoietic organ in teleosts. We dissected kidneys of adult *tbx6:Gal4; Tg(UAS-Cre); A2BD* transgenic animals and observed no contribution of switched DsRed⁺ cells to any hematopoietic lineage (*Figure 8L*). A ubiquitous vascular-specific transgenic Cre driver was utilized as a positive control for hemogenic endothelium, *Tg(kdrl:Cre)^s898* (*Bertrand et al., 2010*; *Figure 8—figure supplement 1*). Taken together, these results demonstrate that SDECs integrate into the dorsal aorta but do not generate HSPCs.

## SDECs support the emergence of HSPCs

Since SDECs did not directly contribute to HSPCs, we wanted to explore their role by analyzing the differential expression of SDECs versus lateral plate mesoderm-derived ECs (LPMDEC) that populated the DA. We compared the SDEC to HE and Pre-HSC clusters at 22–24 hpf. From SDECs, we identified many genes previously attributed to 'niche' functions required to induce aortic hemogenic endothelium (*Figure 9A*; *Charbord et al., 2014*). These results suggested that SDECs might be acting in a paracrine manner to support hematopoietic induction. If so, we reasoned that changes in the number of SDECs might result in the altered formation of hematopoietic cells. Increasing the numbers

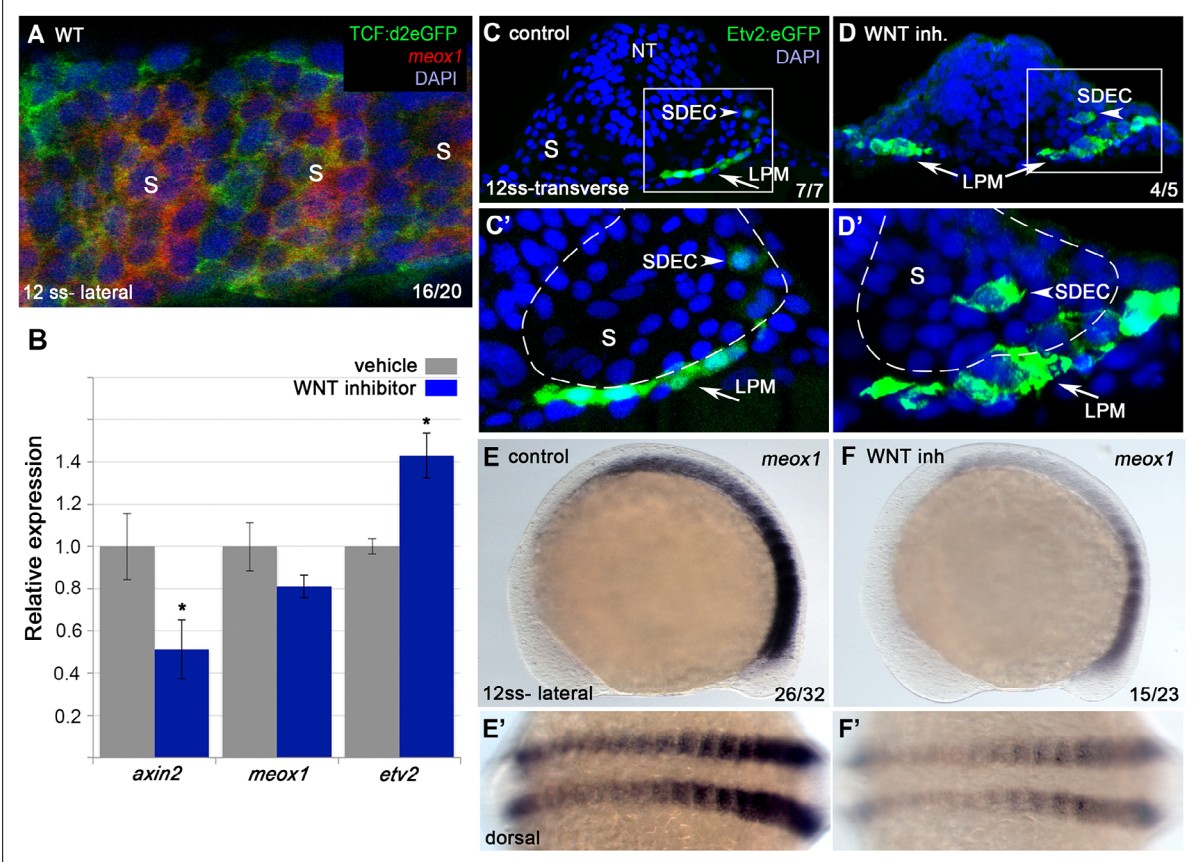

**Figure 7.** Wnt signaling is required for the regionalization of SDECs. (**A**) FISH for *meox1* (red) and antibody staining for a destabilized Wnt/TCF reporter line (green) show co-expression of GFP and *meox1* within the somite. (**B**) Inhibition of Wnt signaling using the chemical inhibitor IWP2 from 2 ss to 15 ss results in decreased expression of *axin2* and *meox1* with a concomitant increase of the expression of *etv2* by qRT-PCR. We observed a reduction in *meox1* expression, although not statistically significant. All genes analyzed between Wnt inhibitor and control embryos, except *meox1*, showed a statistically significant difference (p<0.001, unpaired, two-tailed Student's t-test; n=3.) (**C–D'**) Cross section of *Tg(etv2.1:EGFP)^zf372^* embryos treated with IWP2 from 2 ss to15 ss. *wnt* inhibition results in ectopic formation of Etv2:GFP+ cells within the somite (arrowheads). (**C',D'**) enlargement of somite compartment (dashed lines). Notice LPM cells migrating under the sclerotome (arrows). (**E–F'**) IWP2 control and treated embryos. *wnt* inhibition results in decreased expression of *meox1* by WISH, compared to control embryos. (**E,F**) Lateral view. (**E',F'**) Dorsal view. s, somites; LPM, lateral plate mesoderm; SDECs, somite-derived endothelial cells.

of SDECs via enforced expression of *etv2* with a muscle-specific *mylz2* promoter (*Ju et al., 2003*) or knockdown of *meox1* led to an increase in *runx1*+ by WISH (*Figure 9B–E*). In contrast, decreasing the number of SDECs via over-expression of *meox1* mRNA led to the loss of *runx1*+ and *gata2b*+hematopoietic cells in the DA (*Figure 9F–I*). These results suggest that SDECs act in a paracrine manner to support the induction of hematopoietic cells from neighboring hemogenic endothelial cells.

## Discussion

Here, we present an extensive analysis of how a bipotent skeletal muscle progenitor population gives rise to a rare contingent of ECs that are functionally distinct from endothelial cells specified in the LPM (*Garcia-Martinez and Schoenwolf, 1992*; *Psychoyos and Stern, 1996*; *Schoenwolf et al., 1992*; *Selleck and Stern, 1991*). Through single-cell analysis, we identify unique molecular signatures sufficient to identify subsets of ECs, including somite-derived endothelial cells (SDECs), brain endothelial cells (BVECs), kidney endothelial cells (KVECs), and endocardium as early as the tailbud stage. Together, these results indicate that cell fate restriction of EC subsets occurs earlier than previously thought. Our results show that SDECs emerge from the trunk dermomyotome during somitogenesis and contribute to the dorsal aorta of the zebrafish embryo. We show that modulators of muscle differentiation, such as *meox1*, *notch*, and *wnt*, regulate SDEC number and the timeframe of SDEC

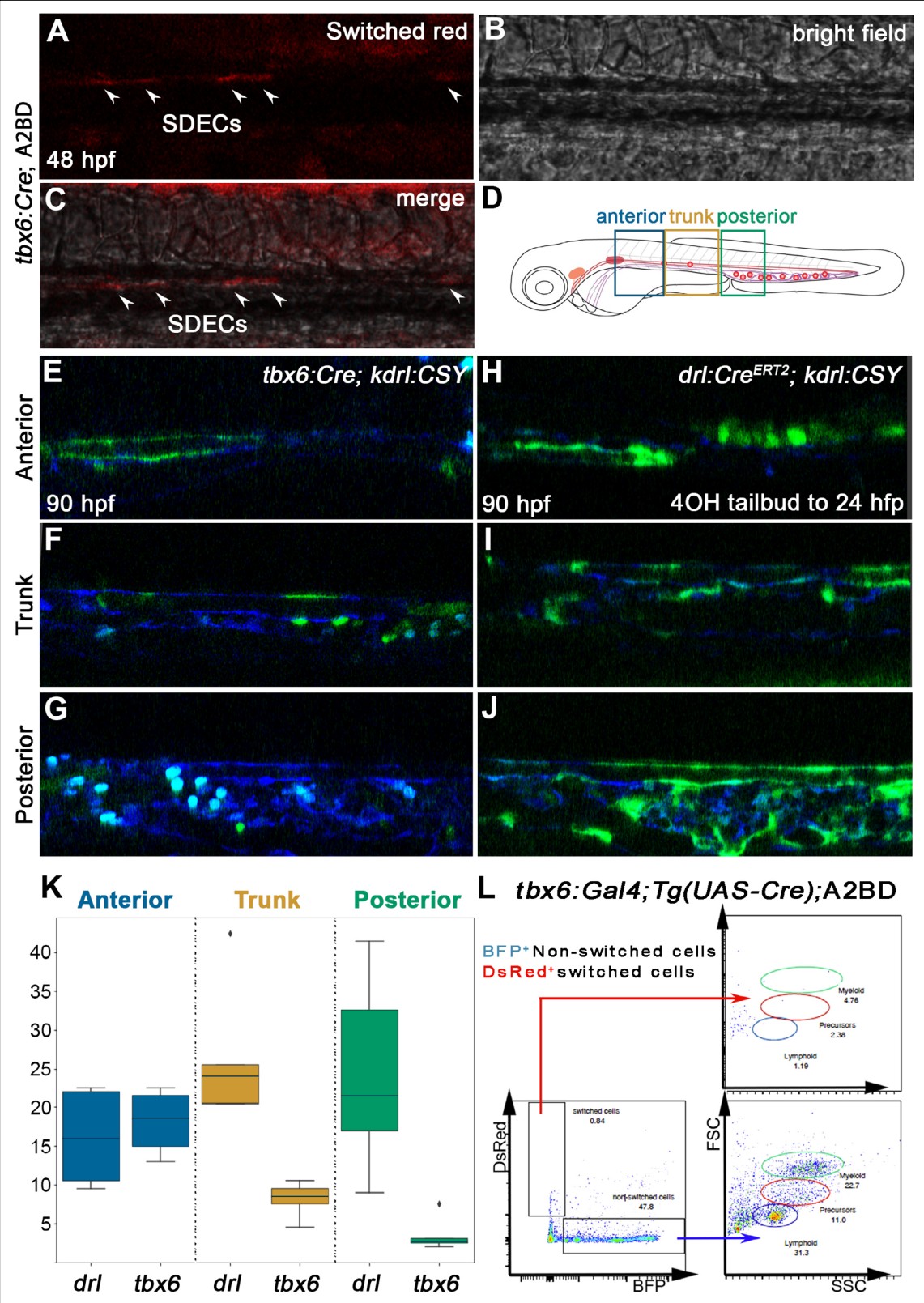

**Figure 8.** SDECs contribute to the dorsal aorta but do not generate HSPCs. (**A–C**) Lineage tracing of SDECs using *tbx6:Gal4; Tg(UAS-Cre); A2BD* shows dsRed+ cells in the vasculature region at 48 hpf (arrowheads). (**E–J**) Using a vasculature-specific switch line *TgBACkdrl:LOXP-AmCyan-LOXP-ZsYellow* (referred to as *kdrl:CSY*), we observe the contribution of SDECs or LPM-derived endothelial cells to the vasculature. (**E–G**) For SDEC labeling, a PM-specific driver *tbx6:Gal4; Tg(UAS-Cre)* was used. PM-derived YFP+ SDECs are observed in the vasculature of imaged embryos. (**H–J**) For LPM-specific

*Figure 8 continued on next page*

*Figure 8 continued*

EC labeling, a *Tg(drl:Cre^{ERT2})* was used and treated with 10 µm tamoxifen starting at 8 hpf. YFP⁺ ECs are observed in all regions of the vasculature. (**K**) Quantification of YFP⁺ SDECs and ECs from *tbx6* or *drl* switched embryos, respectively. Quantifications were based on independent experiments per transgenic background with n=23 for *tbx6* switched embryos and n=9 for *drl* switch embryos. (**L**) Analysis of the adult kidney marrow of *tbx6:Gal4; Tg(UAS-Cre); A2BD* animals shows no contribution to hematopoietic cells from switched DsRed⁺ SDECs through flow cytometry analysis, whereas the FSC/SSC distribution of the unswitched BFP⁺ ECs corresponds to all blood lineages (quantifications based from independent experiments with a total of n=21 samples). SDECs, somite-derived endothelial cells.

The online version of this article includes the following figure supplement(s) for figure 8:

**Figure supplement 1.** Paraxial mesoderm does not generate HSPCs.

---

differentiation. Furthermore, our findings elucidate how SDECs are needed to support the hemogenic program required for the formation of HSPCs.

Avian and murine studies have previously identified a population of endothelial cells that arise from the somites (*Ambler et al., 2001*; *Esner et al., 2006*; *Kardon et al., 2002*; *Mayeuf-Louchart et al., 2016*; *Mayeuf-Louchart et al., 2014*; *Pardanaud et al., 1996*; *Pouget et al., 2006*; *Wilting et al., 1995*; *Yvernogeau et al., 2012*). Previously, the common belief was that in zebrafish, all endothelial and blood cells arise exclusively from the LPM (*Childs et al., 2002*; *Jin et al., 2005*; *Kohli et al., 2013*; *Lawson and Weinstein, 2002*; *Zhang and Rodaway, 2007*). Furthermore, angioblasts are thought to be specified as an equipotent population that undergoes progressive cell-fate restriction only upon migration to the midline (*Kobayashi et al., 2014*; *Lawson et al., 2001*). To our knowledge, only two studies have shown that paraxial mesoderm can generate endothelial cells in zebrafish (*Martin and Kimelman, 2012*; *Nguyen et al., 2014*). The first noted that presomitic mesoderm could generate ECs that incorporate into caudal blood vessels following early inhibition of the Wnt signaling pathway after gastrulation (*Martin and Kimelman, 2012*). The second study highlighted a rare population of ECs generated in the somite that migrated to the developing dorsal aorta (*Nguyen et al., 2014*). Each of these studies was based on retrospective analyses, making it difficult to ascertain precisely when and where SDECs arise and the extent of their contribution. In addition, studies by *Murayama et al., 2023*; *Murayama et al., 2015* have shown that a population of stromal cells important in the maintenance of zebrafish HSPCs derive from the sclerotome, the ventrolateral domain of the somite. These do not appear to give rise to ECs but rather to stromal cells that reside adjacent to blood vessels. As we have occasionally observed rare cells outside of the vasculature in our lineage-tracing studies, some of our marked cells may include this stromal cell subset. Further studies are required to ascertain how this population of somite-derived cells compares to SDECs in the trophic support of hematopoietic precursors.

Labeling individual somite pairs using *Tg(actb2:nls-Eos)*, a pan-nuclear photoconvertible transgenic line, allowed us to quantify the contribution of SDECs from trunk somites. Using *Tg(etv2.1:eGFP)^{zf372}*, with *etv2* as one of the earliest markers of endothelial cell fate acquisition, allowed us to visualize precisely when and where SDECs emerge. Lastly, we could permanently and differentially mark ECs from the LPM or PM using an endothelial-specific switch transgene and follow their respective contributions to the vasculature. Together, both temporal and indelible lineage tracing approaches showed that somites 5–18 have potential to generate SDECs. Emergence of SDECs initiated at the 12-somite stage, concomitant with the epithelization of the somite and the migration of LPM-derived ECs to the midline. Notably, contribution of SDECs to adult hematopoietic tissues was not observed, consistent with many previous studies concluding that the adult hematopoietic program is LPM-derived (*Henninger et al., 2017*; *Jin et al., 2007*; *Murayama et al., 2006*). These results are consistent with a previous study by *Nguyen et al., 2014* that identified a rare population of EC precursors born in a central location within somite, which they termed the 'endotome', that incorporated into the axial vessels but did not appear to generate blood (*Nguyen et al., 2014*). By contrast, in our study, we observed that somite-derived *etv2*⁺ ECs arise from the hypaxial dermomyotome, which is consistent with earlier observations in avian and mouse embryos (*Eichmann et al., 1993*; *Ema et al., 2006*; *Pouget et al., 2006*; *Tozer et al., 2007*). Notably, the hypaxial dermomyotome comprises bipotent-skeletal muscle progenitors that generate ECs or myoblasts upon local cues. Since these bipotent progenitors are present in the territory where the hypaxial muscle precursors originate (*Christ and Ordahl, 1995*), we believe a more appropriate term for this somitic compartment would be the 'myo-endotome'.

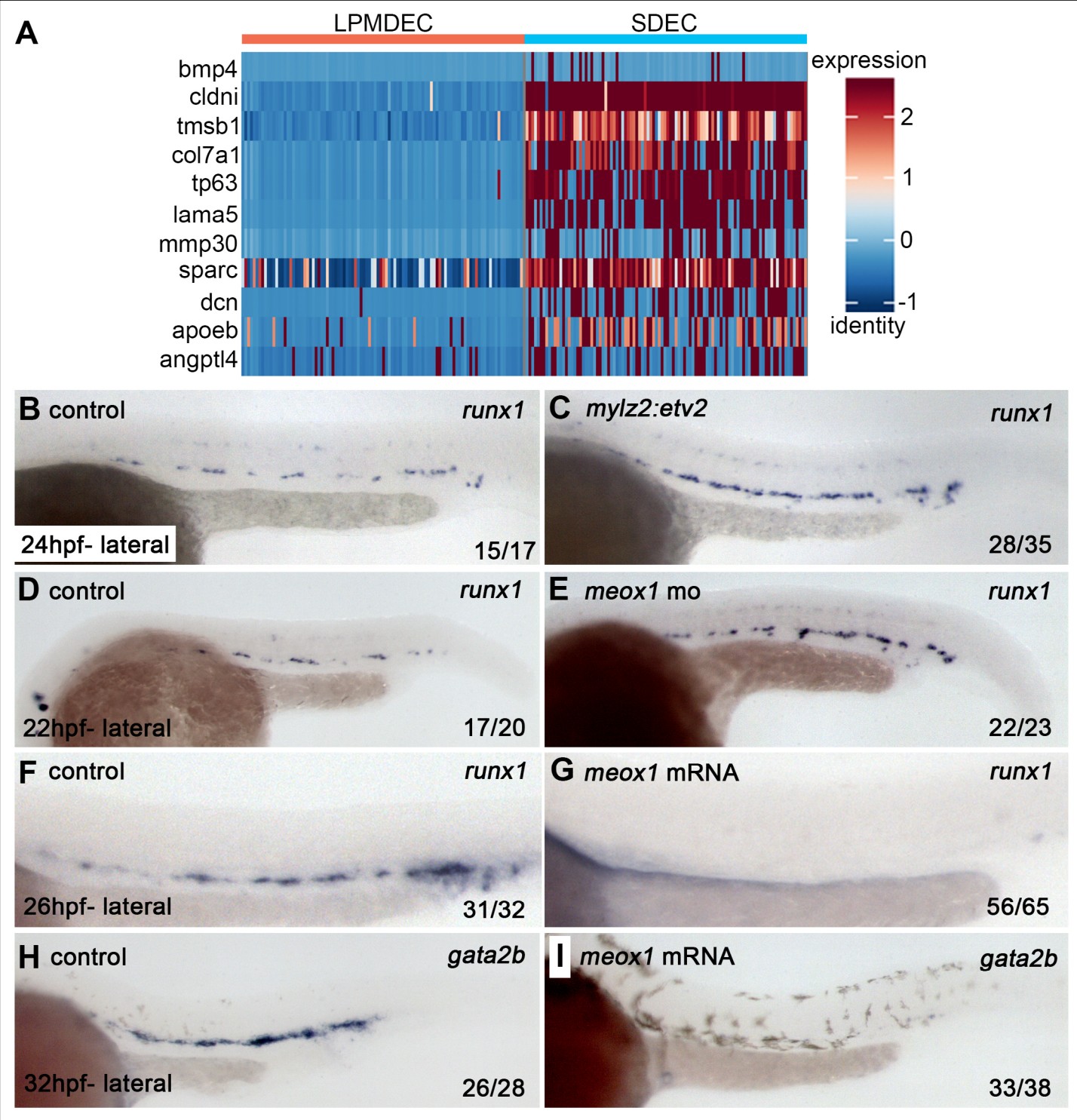

**Figure 9.** SDECs act as a vascular niche for hemogenic endothelium. (**A**) Heatmap of genes differentially regulated between LPM-derived endothelial cells (LPMDEC sample is composed of pre-HSCs and HE clusters) and SDECs. (**B,C**) Zebrafish embryos were injected with an expression vector containing a somite-specific promoter, *mylz2*, driving an *etv2* transgene to ectopically induced SDECs. By WISH, an increase in *runx1* expression was observed at 24 hpf compared to embryos injected with an empty *mylz2* vector. (**D,E**) Similarly, *meox1* morphants exhibit an increased expression of *runx1* by WISH compared to uninjected control embryos. (**F–I**) Conversely, overexpression of *meox1* by mRNA injection strongly reduces the hemogenic markers *runx1* and *gata2b*. SDECs, somite-derived endothelial cells.

To better understand the signaling pathways regulating SDEC emergence, we tested the roles of *notch*, *wnt*, and *npas4l*. We find that Notch signaling is essential for the maintenance of muscle progenitors, as previously shown in mice (*Schuster-Gossler et al., 2007*). Loss of the Notch signaling components *mib* or *notch3* resulted in a premature depletion of skeletal muscle progenitor markers and a concomitant increase of differentiated muscle and endothelial cell gene expression. Our results thus suggest that Notch signaling is required to maintain muscle progenitors rather than determining cell fate choice between muscle and endothelium. However, further work is needed to investigate whether the reduced expression of skeletal muscle progenitor markers results from progenitor cell depletion or simply reflects lower expression levels within these cells. In addition, Wnt signaling is necessary for the regionalization of muscle and endothelial cells within the somite, in agreement with previous work (*Borello et al., 2006*; *Martin and Kimelman, 2012*). We find that during homeostasis, a few *etv2:eGFP*⁺ delaminate from the hypaxial dermomyotome. However, combined loss of Notch signaling and *meox1* shows that EC competence within the somite encompasses a much larger territory, extending as far as the median dermomyotome. Interestingly, we find that endothelial potential within bipotent skeletal muscle progenitors is *npas4l*-dependent. These results are surprising since *npas4l* has been shown to be the earliest gene required for EC specification and was thought to be restricted to the LPM starting at the tailbud stage (*Reischauer et al., 2016*). Equally surprising, a recent paper demonstrated that ECs populating the zebrafish caudal hematopoietic tissue are endoderm-derived (*Nakajima et al., 2023*). Similar to SDECs, this novel population is likewise dependent upon *npas4l* function (*Nakajima et al., 2023*). Together, these results suggest that *npas4l* is required to generate endothelial cell fate regardless of tissue origins.

Recent studies have elucidated a role for non-hemogenic ECs in supporting the hemogenic program and in amplifying HSPC number in various systems (*Butler et al., 2012*; *Butler et al., 2010*; *Butler and Rafii, 2012*; *Gori et al., 2017*; *Gori et al., 2015*; *Guo et al., 2017*; *Hadland et al., 2015*; *Kim et al., 2014*; *Kobayashi et al., 2010*; *Lis et al., 2017*; *Raynaud et al., 2013*; *Sandler et al., 2014*). Interestingly, several groups have successfully differentiated endothelial precursors into transplantable HSPCs, a process that requires an endothelial 'niche' population (*Lis et al., 2017*; *Sandler et al., 2014*). Our work indicates that changes to the number of SDECs via downregulation or overexpression of *meox1* resulted in increased or decreased hematopoietic factors, such as *runx1* and *gata2b*, respectively. This finding and evidence that SDECs lack hemogenic potential suggest that SDECs are important as niche support cells to induce HSPC formation. Future work will elucidate the molecular nature of this interaction and will forward efforts to instruct HSPC fate from human pluripotent precursors.

## Materials and methods
### Zebrafish husbandry
Wild-type AB* and transgenic *TgBAC(etv2:Kaede)^ci6* (*Kohli et al., 2013*) referred to as *etv2:Kaede*, *Tg(fli1:DsRed)^um13* (*Villefranc et al., 2007*) referred to as *fli1:DsRed*, *Tg(tp1:GFP)^um14* (*Parsons et al., 2009*) referred to as *tp1:GFP*, *Tg(drl:H2B-dendra)* (*Mosimann et al., 2015*) referred to as *drl:H2B-dendra*, *Tg(actb2:nls-Eos)* (*Cruz et al., 2015*) referred to as *actb2:nls-Eos*, *Tg(fli1:eGFP)^y1* (*Isogai et al., 2003*) referred to as *fli1:eGFP*, *Tg(etv2.1:eGFP)^zf372* (*Veldman and Lin, 2012*) referred to as *etv2:eGFP*, *Tg(phlbd1:Gal4-mCherry)* (*Distel et al., 2009*) referred to as *phlbd1:mCherry*, *notch3^fh332/fh332* (*Alunni et al., 2013*) referred to as *notch3^-/-*, *Clo^s9/s9* (*Reischauer et al., 2016*) referred to as *npas4l^-/-* (*cloche*), *Tg(7xTCF-Xla.Siam:GFP)^ia4* (*Moro et al., 2012*) referred to as *TCF:d2GFP*, *Tg(tbx-6:Gal4FF:GFP-nls)* (*Yabe et al., 2016*) referred to as *tbx6:Gal4*; *Tg(UAS-Cre)** (*Butko et al., 2015*) referred to as *UAS-Cre*, *Tg(actb2:loxP-BFP-loxP-DsRed)^sd27* (*Kobayashi et al., 2014*) referred to as *A2BD*, *TgBAC(kdrl:LOXP-AmCyan-LOXP-ZsYellow)* (*Zhou et al., 2011*) referred to as *kdrl:CSY*, *Tg(drl:Cre^ERT2)* (*Henninger et al., 2017*) referred to as *drl:Cre^ERT2*, and *Tg(kdrl:Cre)^s898* (*Bertrand et al., 2010*) referred to as *kdrl:Cre*. Zebrafish embryos and adult fish were raised in a circulating aquarium system (Aquaneering) at 28 °C and maintained in accordance with UCSD Institutional Animal Care and Use Committee (IACUC) guidelines (Protocol number: S04168), the Association for Assessment and Accreditation of Laboratory Animal Care International (AAALAC) (Accreditation number: 000503), and the Public Health Service (PHS) Policy on Humane Care and Use of Laboratory Animals (Assurance

number: A3033-1 or D16-00020). All experiments were performed under approved methods of anesthesia, and every effort was made to minimize suffering.

## FACS

*TgBAC(etv2:Kaede)^ci6* embryos were collected at 15 ss and 22 hpf. *Tg(drl:H2B-dendra)* embryos were collected at tailbud stage, 12 ss, and 22 hpf. *Tg(fli1:DsRed)^um13*; *Tg(tp1:GFP)^um14* embryos were collected at 24 hpf. Samples were separately dissociated in phosphate-buffered saline (PBS) supplemented with 10% fetal bovine serum (FBS) by homogenizing them with a sterile plastic pestle or pipette. Dissociated cells were filtered through a 35 µm nylon cell strainer (Falcon 2340) and then rinsed with PBS with 10% FBS. Propidium iodide (Sigma) was added (1 µg ml$^{-1}$) to exclude dead cells and debris. FACS was performed using GFP, Dendra, and DsRed fluorescence gating with a FACS Aria II flow cytometer (Beckton Dickinson).

## Single-cell RNA sample preparation

After FACS, total cell concentration and viability were ascertained using a TC20 Automated Cell Counter (Bio-Rad). Samples were resuspended in 1XPBS with 10% bovine serum albumin (BSA) at a concentration between $8E^2$-$3E^3$ cells per ml. Samples were loaded on the 10 X Chromium system and processed as per manufacturer's instructions (10 X Genomics). Single-cell libraries were prepared per the manufacturer's instructions using the Single Cell 3' Reagent Kit v2 (10 X Genomics). Single-cell RNA-seq libraries and barcode amplicons were sequenced on an Illumina HiSeq2500 platform.

## Single-cell RNA sequencing analysis

The Chromium 3' sequencing libraries were generated using Chromium Single Cell 3' Chip kit v3 and sequenced with an Illumina HiSeq2500 platform. The Ilumina FASTQ files were used to generate filtered matrices using CellRanger 3.0.0 (10X Genomics) with default parameters and imported into R for exploration and statistical analysis using a Seurat package V4 (*La Manno et al., 2018*). Counts were normalized according to total expression, multiplied by a scale factor (10,000), and log-transformed. For cell cluster identification and visualization, gene expression values were also scaled according to highly variable genes after controlling for unwanted variation generated by sample identity. Cell clusters were identified based on UMAP of the first 20 principal components of PCA using Seurat's method, Find Clusters, with an original Louvain algorithm and resolution parameter value 0.5. To find cluster marker genes, Seurat's method, FindAllMarkers was used. Only genes exhibiting a significant p-value of (<0.022), and a minimal average absolute log2-fold change of 1 between each cluster and the rest of the dataset, were considered differentially expressed. To merge individual 22–24 hpf datasets and remove batch effects, Seurat v3 Integration and Label Transfer standard workflow (*Stuart et al., 2019*) was used. To project samples of embryonic stage <22 hpf onto the reference 22–24 hpf dataset, Seurat's MapQuery function was used.

## UV-driven photoconversions

*Tg(actb2:nls-Eos); Tg(fli1:eGFP)^y1* embryos were collected and staged at developmental stages ranging from 4 to 18 ss in a glass bottom Petri dish that was precast using a 2% low melting agarose (Fisher Bioreagents, BP160-500) and a custom-made stamp (Idylle-lab) to created semi-spherical wells. Embryos were then staged and photoconverted with newly emerging somites directly at the bottom of the dish using a Leica SP8 confocal microscope. Recently developed somite pairs were selected by setting a region of interest, then photoconverted by applying maximum UV exposure (405 nm; 100% laser power) using a 20 x objective lens and high digital zoom (>X4) for 90 s. To validate the quality and specificity of the somite photoconversion, images were taken of each channel, eGFP (488 nm; 10% laser power) and mCherry (564 nm; 25% laser power). Photoconverted zebrafish embryos were then transferred to a petri dish with 1-phenyl 2-thiourea (PTU) media and incubated at 28.5 °C in the dark. At 32 hpf, embryos were embedded in a glass bottom petri dish covered by 1% agarose and imaged on a Leica SP8 confocal microscope using resonant scanning. Images of the dorsal aorta were taken in a sequential stack manner for eGFP (488 nm; 10% laser power) and mCherry (564 nm; 35% laser power) using a 20 x objective lens.

## Quantification of SDECs emerging from somites

Maximum intensity projection (MIP) images were taken for each converted embryo to identify the converted somite pair. Then, SDECS were identified and quantified in the dorsal aorta by examining

individual z-stacks and visualizing colocalization of *fli1+*; *actb2:nlsEos^RFP* converted cells in *Tg(actb2:n-ls-Eos)*; *Tg(fli1:eGFP)^y1* embryos. Particularly in *fli1:eGFP-* embryos, we identified SDECs by observing *actb2:nlsEos^Red* cells on the floor of the DA based on the brightfield channel. Statistical analysis of the number of SDECs per somite was performed using Prism 9 (GraphPad).

## Statistical analysis and graphic display

Graphic data representation and statistical analysis were performed using Prism 9 (Graphpad Version 9.5.1). In *Figure 3*, All graphs with single dot scatter plots indicate the mean as column height, and standard error of the mean (SEM) is represented as error bars. In *Figures 5–7*, the statistics of the qPCR were performed using unpaired, two-tailed t-tests (n=3), and standard error of the mean (SEM) is represented as error bars.

## Microscopy and timelapse videos

Embryos were embedded and sectioned as described by *Kobayashi et al., 2014*. Live transgenic embryos and flat-mount or whole-mount two-color double FISH samples were imaged using confocal microscopy (SP5 or SP8, Leica). Since the emission spectra of CFP and YFP overlap, images were captured in two separate sequences to filter overlapping signals by limiting the PMT detection. For CFP a 476 nm laser was used, and the PMT detector was set to collect signals ranging between 480 nm and 505 nm. For YFP, a 514 nm laser was used, and the PMT detector was set to collect signals ranging between 520 nm and 570 nm.

## Microinjections of morpholinos, mRNA, and plasmids

Antisense morpholinos (MOs; Gene Tools, LLC) were diluted as 1- or 3 mM stock in $H_2O$. *meox1*-MO (CTGGCTGACTGTTCCATACTGAAGA) and *mib*-MO (GCAGCCTCACCTGTAGGCGCACTGT) were injected at the 1–2 cell stage of development. *mOrange-CAAX* mRNA was synthesized from linearized mOrange-CAAX with the mMessege mMaching kit (Ambion). 100 pg *mOrange-CAAX* mRNA was injected into one- to two-cell stage embryos. A total of 25 pg of *mylz2:etv2* construct was coinjected with 50 pg of *transposase* mRNA. All microinjections were performed with the indicated RNA or MO concentration in a 1 nl using a PM 1000 cell microinjector (MDI).

## WISH

Whole-mount single or double enzymatic in situ hybridization was performed on embryos fixed overnight with 4% paraformaldehyde (PFA) in PBS. Fixed embryos were washed briefly in PBS and transferred into methanol for storage at −20 °C. Embryos were rehydrated stepwise through methanol in PBS +0.1% Tween 20 (PBT). Rehydrated embryo samples were then incubated with 10 µg ml⁻¹ proteinase K in PBT for 5 min for 5–10 somite stage (12–15 hpf) embryos and 15 min for 24–36 hpf embryos. After proteinase K treatment, samples were washed in PBT and refixed in 4% PFA for 20 min at room temperature. After washes in two changes of PBT, embryos were prehybridized at 65 °C for 1 hr in hybridization buffer (50% formamide, 5 x saline-sodium citrate (SSC), 500 µg ml⁻¹ torula tRNA, 50 µg ml⁻¹ heparin, 0.1% Tween 20, 9 mM citric acid (pH 6.5)). Samples were then hybridized overnight in hybridization buffer including digoxigenin (DIG)- or fluorescein-labeled RNA probe. After hybridization, experimental samples were washed stepwise at 65 °C for 15 min each in hybridization buffer in 2×SSC mix (75%, 50%, 25%), followed by two washes with 0.2×SSC for 30 min each at 65 °C. Further washes were performed at room temperature for 5 min each with 0.2×SSC in PBT (75%, 50%, 25%). Samples were incubated in PBT with 2% heat-inactivated goat serum and 2 mg ml⁻¹ BSA (block solution) for 1 hr and then incubated overnight at 4 °C in block solution with diluted DIG-antibodies (1:5,000) conjugated with alkaline phosphatase (AP) (Roche). To visualize WISH signal, samples were washed three times in AP reaction buffer (100 mM Tris, pH 9.5, 50 mM MgCl₂, 100 mM NaCl, and 0.1% Tween 20) for 5 min each and then incubated in the AP reaction buffer with NBT/BCIP substrate (Roche).

For two-color double FISH, embryos were blocked in maleic acid buffer (MAB; 150 mM maleic acid, 100 mM NaCl, pH 7.5) with 2% Roche blocking reagent (MABB) for 1 hr at room temperature, after hybridizing at 65 °C with probes as described above. Embryos were incubated overnight at 4 °C in MABB with anti-fluorescein POD. (Roche) at a 1:500 dilution. After four washes in MABB for 20 min, each followed by washes in PBS at room temperature, embryo samples were incubated in TSA Plus

Fluorescein Solution (Perkin Elmer) for 1 hr. Embryos were washed for 10 min each in methanol in PBS (25%, 50%, 75%, 100%). Embryos were incubated in 1% $H_2O_2$ in methanol for 30 min at room temperature and washed for 10 min each in methanol in PBS (75%, 50%, 25%) and 10 min in PBS. After blocking for 1 hr in MABB, embryos were incubated overnight at 4 °C in MABB with anti-DIG POD. (Roche) at a 1:1000 dilution. As described above, samples were washed and incubated in TSA Plus CY3 solution (Perkin Elmer). Embryos were washed thrice for 10 min each in PBT and refixed in 4% PFA after the complete staining. Antisense RNA probes for the following genes were prepared using probes containing DIG or fluorescein-labeled UTP: *etv2* (*Clements et al., 2011*), *pax3a* (*Minchin et al., 2013*), *myod* (*Minchin et al., 2013*), *meox1* (*Nguyen et al., 2014*), *runx1* (*Burns et al., 2005*), and *gata2b* (*Butko et al., 2015*).

## Quantitative real-time RT-PCR

Total RNA was collected from whole embryos (~20 embryos) using TRIzol reagent (Ambion, Life Technologies) and isolated from *notch3*^fh332/fh332 mutant embryos or control siblings at 48 hpf; *cloche* mutant embryos or control siblings at 48 hpf; or IWP2 treated and control embryos at 15 hpf with the RNeasy kit (Qiagen). cDNA was generated from total RNA with iScript cDNA synthesis kit (Bio-Rad). The following primers were used for cDNA quantification: *ef1α* (forward, 5'- GAGAAGTTCGAGAAGG AAGC –3'; reverse, 5'- CGTAGTATTTGCTGGTCTCG –3'), *etv2* (forward, 5'- CGAGGTTCTGGTAGGT TTGAG –3'; reverse, 5'- GCACAAAGGTCATGTTCTCAC –3'), *fli1a* (forward, 5'- CGTCAAGCGAGA GTATGACC –3'; reverse, 5'- AGTTCATCTGAGACGCTTCG –3'), *myod1* (forward, 5'- GAAGACGG AACAGCTATGAC –3'; reverse, 5'- GGAGTCTCTGTGGAAATTCG –3'), *myf5* (forward, 5'- CCAGACAG TCCAAACAACAGACC –3'; reverse, 5'- TGAGCAAGCAGTGTGAGTAAGCG –3'), *pax3a* (forward, 5'- ATTCCTTGGAGGTCTCTACG –3'; reverse, 5'- CTACTATCTTGTGGCGGATG –3'), *pax7b* (forward, 5'- CAGTATTGACGGCATTCTGGGAG –3'; reverse, 5'- TCTCTGCTTTCTCTTGAGCGGC –3'), *myf5* (forward, 5'- GAATAGCTACAACTTTGACG –3'; reverse, 5'- GTAAACTGGTCTGTTGTTTG –3'), *myog* (forward, 5'- GTGGACAGCATAACGGGAACAG –3'; reverse, 5'-TCTGAAGGTAACGGTGAGTCGG-3').

## Immunofluorescence

Whole-mount immunofluorescence staining for *myoHII* was performed on *notch3*^fh332/fh332 embryos at 48 hpf using a MF-20 (DSHB) antibody. Immunofluorescence was performed as previously described in *Alexander et al., 1998*.

## Pharmacological treatment

IWR (Tocris) was dissolved in DMSO at a concentration of 10 mM. Zebrafish AB* embryos were incubated in 10 µM IWR solution from 11 to 15 hpf, then fixed with 4% PFA or snap frozen for RNA collection.

## Analysis of whole kidney marrow cells

Whole kidney marrow cells (WKM) were prepared as previously described in *Kobayashi et al., 2008*. Briefly, adult *tbx6:Gal4; Tg(UAS-Cre); A2BD* or *kdrl:Cre; A2BD* transgenic animals were anesthetized and sacrificed following IACUC regulations. The kidney of individual transgenic animals was dissected then minced with scissors. Hematopoietic cells were mechanically dissociated by trituration in PBS supplemented with 2% of FBS and filtered through 40 µm nylon mesh. Prior to flow cytometry analysis, samples were stained with a viability dye (5 nM of Sytox Red, ThermoFisher) to exclude dead cells from the analysis. Flow cytometry was performed on a FACS Aria II (BD Bioscience). Collected data were analyzed using FlowJo software.

## Plasmid construction

Plasmid expression constructs were generated by standard means using PCR from cDNA libraries generated from zebrafish larvae at 24 hpf. These were cloned into pCS2 +downstream of an SP6 promoter. Previously described *mylz2* promoter (*Ju et al., 2003*) was cloned upstream of *etv2* CDS flanked by Tol2 sites.

## Replicates

All experiments assessing phenotype and expression patterns were replicated in at least two independent experiments. Embryos were collected from independent crosses, and experimental processing

(staining or injections) was carried out on independent occasions. Exceptions to this include the analysis of the kidney hematopoietic cells by flow from *tbx6:Gal4; Tg(UAS-Cre)*; A2BD and the imaging of *tbx6:Gal4; Tg(UAS-Cre); kdrl-CSY* where many embryos were processed, and the corresponding *n* are reported in the associated figure legends. For the analysis of qRT-PCR, three independent experiments were performed per condition. One sequencing run for the single-cell sequencing experiments was performed per timepoint from a pool of at least 100 embryos within the corresponding transgenic background.

## Acknowledgements

We thank members of the Traver laboratory for helpful discussions and Stephanie Grainger for discussions and careful editing of the manuscript. We thank the UCSD Institute for Genomic Medicine sequencing core for supporting the scRNA-seq sample preparation and sequencing. We would like to thank Dr. Peng Guo from the Nikon Imaging Center and Jennifer Santini from the Microscopy core at UCSD for their support. We want to thank Anthony Santella from the Bao laboratory at Memorial Sloan Kettering Cancer Center and Kurt Weiss from the Huisken laboratory at the Morgridge Institute for Research for their support and expert advice. Funding This work was supported by the National Institute of Health [R01-DK074482 to DT and PS-H, TR01-OD026219 to DT]; National Institute of Health [T32-HL086344 to PS-H]; California Institute of Regenerative Medicine [EDUC4-12804 to SE]; American Heart Association [19POST34380328 to OS]; and Ministry of Education of the Czech Republic [RVO: 68378050-KAV-NPUI to OS].

## Additional information

### Funding

| Funder | Grant reference number | Author |
| --- | --- | --- |
| National Institute of Diabetes and Digestive and Kidney Diseases | R01-DK074482 | David Traver<br>Pankaj Sahai-Hernandez |
| NIH Office of the Director | TR01-OD026219 | David Traver |
| National Institutes of Health | T32-HL086344 | Pankaj Sahai-Hernandez |
| California Institute for Regenerative Medicine | EDUC4-12804 | Shai Eyal |
| American Heart Association | 19POST34380328 | Ondrej Svoboda |
| Ministry of Education of the Czech Republic | RVO: 68378050-KAV-NPUI | Ondrej Svoboda |

The funders had no role in study design, data collection and interpretation, or the decision to submit the work for publication.

### Author contributions

Pankaj Sahai-Hernandez, Conceptualization, Data curation, Formal analysis, Validation, Investigation, Visualization, Methodology, Writing – original draft, Writing – review and editing; Claire Pouget, Conceptualization, Formal analysis, Validation, Investigation, Visualization, Methodology, Writing – original draft, Writing – review and editing; Shai Eyal, Conceptualization, Formal analysis, Validation, Investigation, Visualization, Methodology, Writing – original draft, Project administration, Writing – review and editing; Ondrej Svoboda, Data curation, Software, Formal analysis, Validation, Investigation, Visualization, Methodology, Writing – review and editing; Jose Chacon, Formal analysis, Investigation, Writing – review and editing; Lin Grimm, Formal analysis, Validation, Investigation, Writing – review and editing; Tor Gjøen, Software, Formal analysis, Investigation, Writing – review and editing; David Traver, Conceptualization, Supervision, Funding acquisition, Writing – original draft, Writing – review and editing

## Author ORCIDs
Pankaj Sahai-Hernandez ⓘ https://orcid.org/0000-0001-7867-8346
Shai Eyal ⓘ http://orcid.org/0000-0001-8131-8077
David Traver ⓘ http://orcid.org/0000-0002-9652-7653

## Ethics
Zebrafish embryos and adult fish were raised in a circulating aquarium system (Aquaneering) at 28 °C and maintained by UCSD Institutional Animal Care and Use Committee (IACUC) guidelines (Protocol number: S04168), the Association for Assessment and Accreditation of Laboratory Animal Care International (AAALAC) (Accreditation number: 000503), and the Public Health Service (PHS) Policy on Humane Care and Use of Laboratory Animals (Assurance number: A3033-1 or D16-00020). All experiments were performed under approved methods of anesthesia, and every effort was made to minimize suffering.

## Decision letter and Author response
Decision letter https://doi.org/10.7554/eLife.58300.sa1
Author response https://doi.org/10.7554/eLife.58300.sa2

# Additional files

## Supplementary files
• MDAR checklist

## Data availability
All data generated or analysed during this study are included in the manuscript and supporting files. scRNA-seq data are available in ArrayExpress under accession number E-MTAB-13196.

The following datasets were generated:

| Author(s) | Year | Dataset title | Dataset URL | Database and Identifier |
|---|---|---|---|---|
| Sahai-Hernandez P | 2023 | Single Cell Sequencing Data for Endothelial Cell Types | https://doi.org/10.6075/J0GB22J0 | Dryad Digital Repository, 10.6075/J0GB22J0 |
| Svoboda O | 2023 | Single-cell RNA-seq (10x Chromium) of FACS-sorted drl:H2B-Dendra2, etv2:Kaede, fli1:DsRed, and tp1:eGFP transgenic zebrafish embryos | https://www.ebi.ac.uk/arrayexpress/E-MTAB-13196 | ArrayExpress, E-MTAB-13196 |

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
