## [Editor Report]

In this important study, the authors identify an additional source of dorsal aorta endothelium derived from the somites that are conventionally thought of as defined blocks of skeletal muscle. The authors present convincing data that these "somite-derived endothelial cells" (SDECs) arise from bipotential precursors in the dermamyotome that give rise to endothelium or muscle. The authors conclude that these cells support but do not produce emergent hematopoietic stem cells in the dorsal aorta, findings that will be of interest to stem cell and developmental biologists.

---

## [Decision Letter]

**Decision letter after peer review:**

Thank you for sending your article entitled "Dermomyotome-derived endothelial cells migrate to the dorsal aorta to support hematopoietic stem cell emergence" for peer review at *eLife*. Your article is being evaluated by 3 peer reviewers, one of whom is a member of our Board of Reviewing Editors, and the evaluation has been overseen by Didier Stainier as the Senior Editor.

Given the list of essential revisions, including new experiments, the editors and reviewers invite you to respond as soon as you can with an action plan for the completion of the additional work. We expect a revision plan that under normal circumstances can be accomplished within two months, although we understand that in reality revisions will take longer at the moment. We plan to share your responses with the reviewers and then advise further with a formal decision.

In particular, the reviewers had concerns regarding the novelty of some of the major points of the manuscript, such as the derivation and contribution of somite-derived endothelial cells to hematopoietic stem cell development, in light of recent high profile publications. However, they also mentioned several interesting aspects of the work that might be developed further to enhance the suitability for publication. These include but are not limited to further understanding of the role of Notch signaling in the development, migration and function of the cell type of interest, additional transcriptomics profiling at earlier timepoints, as well as the use of alternative lines for expanded lineage tracing studies to further document the functional potential and role of the somite-derived endothelial population. If after reading the comments and suggestions below, you feel that these points can be sufficiently addressed in a reasonable timeframe, we look forward to receipt of a revision plan detailing your plans for the manuscript.

*Reviewer #1:*

The manuscript entitled: Dermomyotome-derived endothelial cells migrate to the dorsal aorta to support hematopoietic stem cell emergence, by Sahai-Hernandez et al. aims to explore the contribution of somite-derived endothelial cells (SDECs) to hematopoietic competence in the embryonic dorsal aorta. This study builds on and significantly extends recent work identifying a paraxial mesoderm derived endothelial population under the regulation of Meox that contributes to hemogenic fate in the dorsal aorta. Here, the authors have examined the interplay of several pathways, including Wnt and Notch, in regard to their contribution to the generation of these somite-derived endothelial cells (SDECs) from specific skeletal muscle progenitors and their subsequent role in HSC formation. The authors find that this population of SDECs, marked by ETV2-reporter expression, is derived specifically from the dermomyotome, in balance with the production of muscle cell types. Consistent with early studies in the chick, they describe integration of the SDEC population into the dorsal aorta itself, and then provide novel insight into their preferential localization as well as an unexpected role in paracrine BMP4 mediated HSC regulation. Interestingly, the authors also show, using the reporter line, that the SDEC population is missing in cloche embryos. Through elegant scRNA-seq analysis and lineage trajectory studies, they identified several transcriptionally distinct endothelial populations in the early embryo, prior to HSC specification and vascular maturation, and show how modulation of Notch signaling, known to be essential for hemogenic fate, does or does not impact their development. Finally, the authors investigate the role of ID1 in Notch associated regulation of HSC development. There are several important novel insights provided by the study, however the flow of the report, and strong initial emphasis on the SDEC population, masks some of the other key observations uncovered regarding endothelial biology and hematopoietic fate in the early vertebrate embryo.

1. The flow of the manuscript is somewhat problematic. The authors perform a very elegant unbiased profiling study which identifies several distinct endothelial populations in the embryo, including the SDECs, at a much earlier timepoint than previously anticipated. Given recent publications, rather than starting with characterization of the SDEC population, and evaluating their production and function with additional precision, the authors might consider inverting the study to describe and then validate the relationships and regulation of the formation of each distinct subset, at least in regard to those impacting blood formation. In essence they end up doing this for SDECs, arterial and hemogenic endothelium already in the later figures, but altering the flow may help to highlight distinctions from prior work, as well as alleviate some of confusion/ redundancy regarding the multiple roles/timing of Notch signaling, and differing mechanistic influences hemogenic fate.

2. With the current layout of the manuscript, the very intriguing studies at the end of the paper regarding ID1 seem disconnected from the main focus on the SDECs. If the authors tell a more generalized story about establishing several distinct populations of endothelial cells that influence HSC formation (it would be fine to almost ignore the few that don't impact HSCs if that is the intended focus), then the ID1 data being associated with LPM-derived endothelium, but not PM-derived cells (in contrast to BMP signaling) is more logical.

3. The authors use an ETV2-reporter to follow the onset of SDEC production as well as their later integration in the aorta. However, they also indicate that ETV2 is expressed before the somite-derived population is present, and the RNAseq analysis seems to indicate its expression is found in many endothelial populations, such that it might not be specific to SDECs. Given those caveats, it would be helpful to find another marker (from the RNAseq?) that could be used to verify some of the data presented. As an earlier study also showed localization of a similar population of somite-associated cells in the dorsal aorta, this conclusion not in doubt, but the reporter could be inclusive of more than one endothelial cell type, impacting specificity of the BMP analysis and/or loss of function/epistasis studies; use of a second marker may help solidify the novel insights.

4. As the dermomyotome is transient, it is interesting to understand how the biology overlaps with HSPC development. It appears the authors presume that the cells of different origin (LPM vs PM) have experienced differential signaling that impacts their later function. The elegant RNAseq data and trajectory analyses imply they are indeed transcriptionally distinct at a later time point. Is it possible to find and compare the hematopoiesis-associated endothelial subsets earlier in development to learn more about the onset of their development and differences, or validate additional distinguishing marks by qPCR or LOF studies prior to integration in the aorta? Given prior reports suggesting a role for CXCL12 in hematopoietic regulation from somite-derived endothelial cells, it would be interesting to know when this signature appears and in which cohort(s).

*Reviewer #2:*

The article by Sahai-Hernandez et al. describes endothelial cells in the somite of developing zebrafish, their delamination to colonize the developing dorsal aorta and a trophic role in aortic hematopoiesis. The paper shows that these cells are derived from a muscle-endothelial precursor present in the lateral lip of the somite starting from the 12-somite stage. Differentiation of these cells are dependent on the Notch or the Wnt pathways Single cell transcriptome approaches have revealed the molecular identity of somitic endothelial cells, in particular the expression of genes related to the hematopoietic support. These single cell studies have been extended to a context of Notch mutant and show that the Notch pathway plays a moderate role on somitic endothelial cells.

This is an interesting article that, unfortunately comes late regarding the paper by Nguyen et al., Nature 2014. A number of data presented here have already been published in the Nature paper in a more convincing way. In particular the dual expression of muscle- and endothelial-specific genes, the muscle/endothelium balance and the trophic role of somitic endothelial cells on aortic hematopoiesis among the most important ones.

In its current form, the article is not acceptable for *eLife* because many of the data and conclusions presented here have already been published in the Nature paper. However, this article brings some novelties that are the timing of appearance and migration of the somite endothelial cells, the single cell characterization and the extensive study of the Notch pathway.

If acceptable by the Editor and feasible by the authors, this reviewer would recommend to significantly reinforce

1. the descriptive aspect of the emergence, migration and integration of cells in the dorsal aorta by insisting on the beginning and the end of the phenomenon, the number of cells per somite, the dynamic aspect of development, Draculin and Tbx6 approaches. In other words, make a kind of atlas for these cells with 3D visualizations if possible.

2. to focus on the Notch pathway. The role of Wnt and Meox have already been published in the Nature paper.

3. to further exploit the single cell transcriptome to show the existence of bipotent precursors and how these precursors enter the endothelial or muscular pathway depending on Notch conditions.

*Reviewer #3:*

This manuscript explores the origins of endothelial cells that make up the dorsal aorta (DA) in the developing zebrafish. The DA is the site of the hemogenic endothelium that produces definitive hematopoietic stem cells (HSCs). The prevailing model is that a subset of lateral plate mesoderm is fated for the dorsal aorta. These cells are further segregated into cells that make up the aortic endothelium, and those with hemogenic potential. In this study the authors identify an additional source of DA endothelium derived from the somites, which are conventionally thought of as defined blocks of skeletal muscle. These cells in the dermamyotome are positive for etv2:gfp+ cells and are proposed as bipotential precursors that can give rise to endothelium or muscle. The authors conclude that these cells are supportive of HSC emergence, but do not themselves produce HSCs. This is a novel concept in the ontogeny of hemogenic endothelium.

1. In Figure 1, it would be clearer if the etv2:gfp line was crossed to a red endothelial-specific line (e.g. flk:mCherry). Without additional landmarks in S1 it is unclear if these cells are continuously migrating towards the midline and becoming incorporated in the DA, or if they are moving out of the focal plane but are still part of the somite. Additionally, the movies S1 and S2 need to have features of interest labeled including boundaries between different tissue types.

2. Following the point above, the migration between etv2:gfp+ cells in the dermamyotome and the DA needs to be better defined. There should be more discrete lineage tracing, such as a photoactivatable marker, or time-lapse live imaging to show incorporation of these cells in the DA. The existing time-lapse shows delamination of etv2:gfp+ cells from the dermamyotome, but not migration and incorporation into the DA.

3. A key conclusion of the study is that SDEC cells are integrated into the DA are supportive of HSCs. If that is the case, the authors could perform time-lapse of tbx6:Cre;kdrl:CSY or some other reporter line combination to observe where budding HSCs are emerging relative to SDECs. This would capture the spatial relationships between the proposed supportive SDECs and HSCs.

4. Does meox1 knockdown result in enhanced proliferation of etv2+ cells in the dermamyotome, or do more cells emerge de novo? Single cell lineage tracing or tracking of single cells in time-lapse movies could help distinguish between these possibilities.

5. In Figures 2O and 3O the change in expression levels from bulk mRNA of pooled whole embryos does not necessarily mean that there is a change in the numbers of specific cell types. This result could also occur if there are similar cell types present that have altered expression profiles. These results could be clarified by FACS analysis of different populations to detect a change in ratios.

6. The authors do not go back further in ontogeny to explore possible heterogeneity within the dermamyotome. For example, how early do the etv2:gfp cells arise from the PSM and can be distinguished as a bipotential dermamyotome precursor?

7. What is the significance or conclusions drawn from the regional anterior-trunk-posterior observations presented in Figure 5?

8. The rationale for choosing lines for scRNA-seq (Etv2:kaede, Fli1:dsRed; Tp1:GFP, and Draculin:Dendra:H2B) is not clear because these lines are not specific to ECs and are known collectively to mark blood, endocardium, and other cell types. How do the authors distinguish between subtypes of ECs versus other cell types altogether?

9. The authors could show in Figure 6 where etv2:kaede and the other sorted reporter lines map onto these clusters. The authors could also provide scRNA-seq data for the same etv2:gfp+ cells that were presented in previous figures to confirm the presence of SDECs.

10. A recent paper comparing a new etv2 knock-in line with existing transgenic lines shows differences in expression patterns (Chestnut & Sumanas Dev Dyn 2020). Although data is presented in this study that shows endogenous expression of etv2 using FISH, it is possible that expression and conclusions drawn from the etv2:gfp line does not precisely follow endogenous cell types. This caveat should be considered and discussed within the larger context of the study.

11. It is unclear from the FISH images if the etv2+ cells are in fact double positive with meox1, or if etv2+ cells are intercalated between meox1 cells. The green meox1 signal appears low within the boundaries of etv2+ cells. Could this be resolved by crossing the etv2:gfp line with a red muscle progenitor reporter line, if available?

12. The authors do not discuss the paper by Murayama et al. (Nat. Comm. 2015) that shows stromal cells in the caudal hematopoietic tissue (CHT) directly delaminate from the somites. There are important similarities between this manuscript and the previous study that are not explored or referenced. For example, in Figure 5G, what are the round cells in the CHT that have switched? Could these be the stromal cells observed by Murayama et al? The switched cells in Figure 5J have endothelial morphology, but not the cells in Figure 5G.

13. The authors say that lineage tracing of tbx6:cre cells in the DA are consistent between multiple embryos over different stages, however this should be better quantified with more of the data represented in the analysis. It would be informative to show data before the 90 hpf timepoint in Figure 5E-J.

14. Are the Cre reporter lines single insertions? It appears from the FACs plots in S3 there are a range of signal intensities, and that some cells may express both colors, suggesting incomplete recombination of multiple insertions. This is possible too in Figure 5G that shows cell that appear double positive. Are there additional controls that could be presented?

15. The draculin-cre result is not surprising as the line markers endothelial and blood precursors that, as expected, are labeled in this experiment. These results are independent of SDEC ontogeny. This should be clarified.

16. In Figure 7A, bmp4 only appears upregulated in a small number of SDECs, and yet the conclusion is made that this is an important factor produced by SDECs.

[Editors' note: further revisions were suggested prior to acceptance, as described below.]

Thank you for resubmitting your work entitled "Dermomyotome-derived endothelial cells migrate to the dorsal aorta to support hematopoietic stem cell emergence" for further consideration by *eLife*. Your revised article has been evaluated by Didier Stainier (Senior Editor) and a Reviewing Editor.

The manuscript has been improved but there are some remaining issues that need to be addressed, as outlined below:

The reviewers are very enthusiastic about the revised manuscript. The only outstanding issue is the presentation of new data that does not have quantification or statistical analysis.

*Reviewer #3 (Recommendations for the authors):*

In this study, the authors identify an additional source of dorsal aorta endothelium derived from the somites, which are conventionally thought of as defined blocks of skeletal muscle. These cells in the dermamyotome are positive for endothelial cell markers and are proposed as bipotential precursors that can give rise to endothelium or muscle. The authors conclude that these cells are supportive of HSC emergence, but do not themselves produce HSCs.

Most of the previous reviewers' comments have been addressed. However, there are new imaging data presented that has not been analyzed quantitatively or rigorously tested for significant changes using statistical tests.

1. Quantification of results in Figure 4F,G, 5A-F, 5I-N, 7C-D.

2. Is decrease in meox1 significant in Figure 7B? If not, this should be clarified.

3. Need more detail about microscopy that allowed resolution of CFP and YFP from kdrl:CSY line in Figure 8E-J.

4. The drl:Cre-ERT2 data in Figure 8H-K is not clearly convincing and could be removed. The tbx6:Cre data makes the point well.

---

## [Author Response]

Essential revisions:Reviewer #1:The manuscript entitled: Dermomyotome-derived endothelial cells migrate to the dorsal aorta to support hematopoietic stem cell emergence, by Sahai-Hernandez et al. aims to explore the contribution of somite-derived endothelial cells (SDECs) to hematopoietic competence in the embryonic dorsal aorta. This study builds on and significantly extends recent work identifying a paraxial mesoderm derived endothelial population under the regulation of Meox that contributes to hemogenic fate in the dorsal aorta. Here, the authors have examined the interplay of several pathways, including Wnt and Notch, in regard to their contribution to the generation of these somite-derived endothelial cells (SDECs) from specific skeletal muscle progenitors and their subsequent role in HSC formation. The authors find that this population of SDECs, marked by ETV2-reporter expression, is derived specifically from the dermomyotome, in balance with the production of muscle cell types. Consistent with early studies in the chick, they describe integration of the SDEC population into the dorsal aorta itself, and then provide novel insight into their preferential localization as well as an unexpected role in paracrine BMP4 mediated HSC regulation. Interestingly, the authors also show, using the reporter line, that the SDEC population is missing in cloche embryos. Through elegant scRNA-seq analysis and lineage trajectory studies, they identified several transcriptionally distinct endothelial populations in the early embryo, prior to HSC specification and vascular maturation, and show how modulation of Notch signaling, known to be essential for hemogenic fate, does or does not impact their development. Finally, the authors investigate the role of ID1 in Notch associated regulation of HSC development. There are several important novel insights provided by the study, however the flow of the report, and strong initial emphasis on the SDEC population, masks some of the other key observations uncovered regarding endothelial biology and hematopoietic fate in the early vertebrate embryo.1. The flow of the manuscript is somewhat problematic. The authors perform a very elegant unbiased profiling study which identifies several distinct endothelial populations in the embryo, including the SDECs, at a much earlier timepoint than previously anticipated. Given recent publications, rather than starting with characterization of the SDEC population, and evaluating their production and function with additional precision, the authors might consider inverting the study to describe and then validate the relationships and regulation of the formation of each distinct subset, at least in regard to those impacting blood formation. In essence they end up doing this for SDECs, arterial and hemogenic endothelium already in the later figures, but altering the flow may help to highlight distinctions from prior work, as well as alleviate some of confusion/ redundancy regarding the multiple roles/timing of Notch signaling, and differing mechanistic influences hemogenic fate.

We thank our reviewer for this excellent suggestion and agree that restructuring the manuscript has provided an emphasis on novelty, a more logical progression, and improved clarity. As suggested, we have now structured the paper to lead with the divergence of the endothelial lineages and the early onset of their specification, as highlighted by the single-cell work. We then focus upon the uniqueness of the somite-derived endothelial cell (SDEC) population and characterize their cellular and molecular nature. We now quantify SDEC formation per somite pair, better characterize their cellular emergence from the somites, characterize the molecular regulation of their emergence, and focus upon their role as niche support cells within the dorsal aorta to regulate the emergence of HSPCs.

2. With the current layout of the manuscript, the very intriguing studies at the end of the paper regarding ID1 seem disconnected from the main focus on the SDECs. If the authors tell a more generalized story about establishing several distinct populations of endothelial cells that influence HSC formation (it would be fine to almost ignore the few that don't impact HSCs if that is the intended focus), then the ID1 data being associated with LPM-derived endothelium, but not PM-derived cells (in contrast to BMP signaling) is more logical.

We agree that in the original layout of the manuscript, the roles of *bmp4* and *id1* seemed out of focus. In the revised manuscript, we have better addressed the specification and contribution of the different EC subsets, followed by a more in-depth characterization of the SDEC population. We have now omitted the section regarding the roles of *id1* and *bmp4* and will address them in another manuscript.

3. The authors use an ETV2-reporter to follow the onset of SDEC production as well as their later integration in the aorta. However, they also indicate that ETV2 is expressed before the somite-derived population is present, and the RNAseq analysis seems to indicate its expression is found in many endothelial populations, such that it might not be specific to SDECs. Given those caveats, it would be helpful to find another marker (from the RNAseq?) that could be used to verify some of the data presented. As an earlier study also showed localization of a similar population of somite-associated cells in the dorsal aorta, this conclusion not in doubt, but the reporter could be inclusive of more than one endothelial cell type, impacting specificity of the BMP analysis and/or loss of function/epistasis studies; use of a second marker may help solidify the novel insights.

*etv2* expression is indeed a marker of commitment to the endothelial lineages and is believed to be expressed by all subsets of ECs. It thus is certainly not a specific marker of SDECs. To specifically mark SDECs, we used a PM-specific lineage tracer, *tbx6:Cre*, and two “switch” lines. In the first combination, *tbx6:Gal4; Tg(UAS-Cre); A2BD* animals show switching of PM-derived lineages from BFP^+^ to dsRed^+^ cells, allowing SDECs to be followed upon leaving the somite and integrating into the DA. In the second combination, *tbx6:Gal4; Tg(UAS-Cre); kdrl:CSY* animals will switch out a CFP cassette to YFP in PM-derived cells. Because this switched transgene will only be expressed in endothelial cells due to the *kdrl* driver, only SDECs will be marked with YFP, whereas all other ECs will remain CFP^+^ due to derivation from non-PM-derived precursors. In both lineage tracing lines, we show PM-derived ECs lodged within the floor of the DA (Figure 8). In addition, we have added a meticulously performed analysis of the migration and localization of SDECs to the DA by using a photoconvertible *Tg(actb2:nls-Eos); Tg(fli1:eGFP)^y1^* transgenic line. By applying precise photoconversion of single somite pairs and following their trajectories into the DA, we further support SDEC contribution to the DA, provide a detailed map of the somites involved (trunk somites), and quantify the number of SDECs contributed by each somite pair (Figure 3). We believe that these additional observations provide further evidence of SDEC emergence, migration, and contribution to the DA. We have not yet identified a single gene product that can be used to specifically identify SDECs.

4. As the dermomyotome is transient, it is interesting to understand how the biology overlaps with HSPC development. It appears the authors presume that the cells of different origin (LPM vs PM) have experienced differential signaling that impacts their later function. The elegant RNAseq data and trajectory analyses imply they are indeed transcriptionally distinct at a later time point. Is it possible to find and compare the hematopoiesis-associated endothelial subsets earlier in development to learn more about the onset of their development and differences, or validate additional distinguishing marks by qPCR or LOF studies prior to integration in the aorta? Given prior reports suggesting a role for CXCL12 in hematopoietic regulation from somite-derived endothelial cells, it would be interesting to know when this signature appears and in which cohort(s).

This is an excellent point. We have now obtained scRNA sequencing data sets from ECs purified at earlier development timepoints. We have added differential expression analyses comparing selected EC subsets between tailbud, 12 ss, 15 ss, and 22 hpf stages. Surprisingly, the tissue-specific signatures we detected at 22 hpf can be traced backward in developmental time. We identified EC clusters with distinct transcriptomes as early as the tailbud stage, suggesting that EC specification starts by the end of gastrulation and the initiation of somitogenesis. While the EC clusters were still heterogeneous prior to the 12 ss stage, we could identify most EC subsets, including HE, pre-HSC, KVECs, BVECs, and SDECs, after the 15 ss when their transcriptomes became more defined. Together, these data suggest a gradual EC specification process that begins as early as the tailbud stage. These findings are highlighted in our revised Figures 1, Figure 1 —figure supplement 1-3, Figure 2, and Figure 2 —figure supplement 1.

Specifically, for the *cxcl12* signaling axis, we have found expression of its components only within early brain ECs, which was downregulated by 22 hpf (Figure 2 —figure supplement 1). The sequencing depth within our scRNA-Seq libraries is limited, which may have precluded us from finding *cxcl12* components within the SDEC subset.

Reviewer #2:The article by Sahai-Hernandez et al. describes endothelial cells in the somite of developing zebrafish, their delamination to colonize the developing dorsal aorta and a trophic role in aortic hematopoiesis. The paper shows that these cells are derived from a muscle-endothelial precursor present in the lateral lip of the somite starting from the 12-somite stage. Differentiation of these cells are dependent on the Notch or the Wnt pathways Single cell transcriptome approaches have revealed the molecular identity of somitic endothelial cells, in particular the expression of genes related to the hematopoietic support. These single cell studies have been extended to a context of Notch mutant and show that the Notch pathway plays a moderate role on somitic endothelial cells.This is an interesting article that, unfortunately comes late regarding the paper by Nguyen et al., Nature 2014. A number of data presented here have already been published in the Nature paper in a more convincing way. In particular the dual expression of muscle- and endothelial-specific genes, the muscle/endothelium balance and the trophic role of somitic endothelial cells on aortic hematopoiesis among the most important ones.In its current form, the article is not acceptable for eLife because many of the data and conclusions presented here have already been published in the Nature paper. However, this article brings some novelties that are the timing of appearance and migration of the somite endothelial cells, the single cell characterization and the extensive study of the Notch pathway.If acceptable by the Editor and feasible by the authors, this reviewer would recommend to significantly reinforce1. the descriptive aspect of the emergence, migration and integration of cells in the dorsal aorta by insisting on the beginning and the end of the phenomenon, the number of cells per somite, the dynamic aspect of development, Draculin and Tbx6 approaches. In other words, make a kind of atlas for these cells with 3D visualizations if possible.2. To focus on the Notch pathway. The role of Wnt and Meox have already been published in the Nature paper.3. To further exploit the single cell transcriptome to show the existence of bipotent precursors and how these precursors enter the endothelial or muscular pathway depending on Notch conditions.

We acknowledge that our study revisits some previous findings from Pete Currie’s group and many seminal studies of SDECs in the chick embryo. We want to point out that our work brings a variety of new insights into the formation and function of SDECs, as outlined above in our response to Reviewer 1. In our revised manuscript, we further these novel findings by improved imaging of SDEC formation and migration from the somite to the aorta by utilizing a photoconvertible *Tg(actb2:nls-Eos); Tg(fli1:eGFP)^y1^* transgenic line (Figure 3). These animals were used to produce a detailed atlas of which somites contribute SDECs to the trunk DA and quantify the range of SDEC contribution per somite pair. Crossing the photoconvertible *Tg(actb2:nls-Eos)* line with a *Tg(fli1:eGFP)^y1^* transgenic line demonstrates how SDECs migrate into the DA. We also utilized several fluorescent transgene combinations, including *tbx6:Cre; kdrl:CSY* and *tbx6:Cre; A2BD*, which demarcates SDECs from LPM-derived ECs (Figure 8).

We extended our studies and discussion on the role of *notch* in SDEC formation and function, as suggested. We show that Notch signaling is dispensable for SDEC fate and is rather required to enforce the muscle cell program from their bipotent precursors. To our knowledge, our results show for the first time the role of Wnt signaling in the muscle vs. endothelial cell fate decision in the dermomyotome by reducing *meox1* expression to yield increased numbers of SDECs. We have thus kept these results in our manuscript. The Wnt signaling referred to in the (Nguyen et al., 2014) paper was specific to *wnt16*, a non-canonical *wnt* ligand that operates exclusively in the sclerotome and thus irrelevant to our current studies.

As for adding more on the bipotency of the early ECs, we have obtained scRNA-Seq data sets from ECs purified at earlier time points in development (Figure 2, and Figure 2 —figure supplement 1). We have now added a differential expression analysis comparing selected EC subsets between 15 ss and 22 hpf. We have found that genetic programs driving endothelial differentiation were upregulated in all subsets tested, in contrast to tissue-specific differentiation programs (i.e., somitogenesis), which were downregulated. These results suggest an early commitment to the EC lineages as early as 15 ss.

Reviewer #3:This manuscript explores the origins of endothelial cells that make up the dorsal aorta (DA) in the developing zebrafish. The DA is the site of the hemogenic endothelium that produces definitive hematopoietic stem cells (HSCs). The prevailing model is that a subset of lateral plate mesoderm is fated for the dorsal aorta. These cells are further segregated into cells that make up the aortic endothelium, and those with hemogenic potential. In this study the authors identify an additional source of DA endothelium derived from the somites, which are conventionally thought of as defined blocks of skeletal muscle. These cells in the dermamyotome are positive for etv2:gfp+ cells and are proposed as bipotential precursors that can give rise to endothelium or muscle. The authors conclude that these cells are supportive of HSC emergence, but do not themselves produce HSCs. This is a novel concept in the ontogeny of hemogenic endothelium.1. In Figure 1, it would be clearer if the etv2:gfp line was crossed to a red endothelial-specific line (e.g. flk:mCherry). Without additional landmarks in S1 it is unclear if these cells are continuously migrating towards the midline and becoming incorporated in the DA, or if they are moving out of the focal plane but are still part of the somite. Additionally, the movies S1 and S2 need to have features of interest labeled including boundaries between different tissue types.

We agree that our previous figures could have been more clear. We have now overhauled our imaging figures and present improved timelapse imaging movies. To better target the somites and image their contribution, we lineage traced PM-specific ECs, and photoconverted somite-derived cells (see explanation below). In addition, we annotated features and boundaries in Figure 4 – video 1 and 2.

2. Following the point above, the migration between etv2:gfp+ cells in the dermamyotome and the DA needs to be better defined. There should be more discrete lineage tracing, such as a photoactivatable marker, or time-lapse live imaging to show incorporation of these cells in the DA. The existing time-lapse shows delamination of etv2:gfp+ cells from the dermamyotome, but not migration and incorporation into the DA.

Agreed. We now present much improved lineage tracing experiments. To specifically mark SDECs, we used a PM-specific lineage tracer, *tbx6:Cre*, and two “switch” lines. In the first combination, *tbx6:Gal4; Tg(UAS-Cre); A2BD* animals show switching of PM-derived lineages from BFP^+^ to DsRed^+^ cells, allowing SDECs to be followed upon leaving the somite and integrating into the DA. In the second combination, *tbx6:Gal4; Tg(UAS-Cre); kdrl:CSY* animals will switch out a CFP cassette to YFP in PM-derived cells. Because this switched transgene will only be expressed in endothelial cells due to the *kdrl* driver, only SDECs will be marked with YFP, whereas all other ECs will remain CFP^+^ due to derivation from non-PM-derived precursors. In both lineage tracing lines, we show that the vast majority of PM-derived ECs migrate to the floor of the DA. In addition, we have added a meticulously performed analysis of the migration and localization of SDECs to the DA by using a photoconvertible *Tg(actb2:nls-Eos); Tg(fli1:eGFP)^y1^* transgenic line. By applying precise photoconversion of single somite pairs and following their trajectories into the DA, we further support the contribution of SDECs to the DA, provide a detailed map of the somites involved (trunk somites), and quantify the number of SDECs contributed by each somite pair.

3. A key conclusion of the study is that SDEC cells are integrated into the DA are supportive of HSCs. If that is the case, the authors could perform time-lapse of tbx6:Cre;kdrl:CSY or some other reporter line combination to observe where budding HSCs are emerging relative to SDECs. This would capture the spatial relationships between the proposed supportive SDECs and HSCs.

See response to comment 2. Capturing the precise interplay over time between SDECs and HE within the DA is difficult and requires generation of new transgenic lines. We are currently building these as we feel this is an exciting avenue for our future studies.

4. Does meox1 knockdown result in enhanced proliferation of etv2+ cells in the dermamyotome, or do more cells emerge de novo? Single cell lineage tracing or tracking of single cells in time-lapse movies could help distinguish between these possibilities.

This issue was addressed in the studies of (Nguyen et al., 2014), where they performed phosphohistone H3 staining in *meox1* mutants. They showed no increase in proliferation within the somites in *meox1* mutants, suggesting that the increase in SDECs is likely due to a difference in cell fate outcomes. Our data support this postulate in that further genetic perturbations (in *notch3*^-/-^; *meox1* KD animals, e.g.) reveal broad endothelial potential within the somite. Ectopic EC formation appears to occur via lineage skewing within bipotent muscle / EC precursors within the dermomyotome.

5. In Figures 2O and 3O the change in expression levels from bulk mRNA of pooled whole embryos does not necessarily mean that there is a change in the numbers of specific cell types. This result could also occur if there are similar cell types present that have altered expression profiles. These results could be clarified by FACS analysis of different populations to detect a change in ratios.

We agree that the results from bulk mRNA of pooled whole embryos in *notch3* and *cloche* mutants are suggestive and do not necessarily indicate a change in cell number. FACS analysis for these different populations in their respective mutant backgrounds is not feasible due to the crosses required. Instead, we toned down our conclusions and suggested further inquiry into the idea that loss of *notch3* results in premature depletion of muscle progenitors or reduced expression/activity within these cells.

6. The authors do not go back further in ontogeny to explore possible heterogeneity within the dermamyotome. For example, how early do the etv2:gfp cells arise from the PSM and can be distinguished as a bipotential dermamyotome precursor?

We have obtained scRNA-Seq data sets from ECs purified at earlier development times (Figure 2, and Figure 2 —figure supplement 1). We show that specification of EC subsets, including SDECs, can already be observed as early as tailbud stage. As noted above in response to Reviewer 1, by comparing selected EC subsets between 15 ss and 22 hpf, we have found evidence suggesting an early commitment to the EC lineages following the end of gastrulation. Similar to observations in the avian model (Pouget et al., 2006), we have never detected endothelial transcripts or Etv2:GFP^+^ cells in the PSM. In addition, the earliest Etv2:GFP^+^ cells were observed at 12 ss. We also confirmed this observation by FISH, where we observed *etv2^+^ / meox1^+^* and *etv2^+^ / pax3^+^* cells at 12 ss. Furthermore, when we knocked down *meox1*, we observed ectopic formation of *m*eox1^+^ / *etv2^+^* cells. Together, these data indicate that SDECs arise from a bipotent progenitor population as early as the 12ss.

7. What is the significance or conclusions drawn from the regional anterior-trunk-posterior observations presented in Figure 5?

That, relative to cells marked by a *drl:Cre*^ERT2^ lineage tracer that labels all ECs, a *tbx6:Cre* driver specifically targeting SDECs generates ECs that traffic only to the anterior / trunk portions of the aorta. We have clarified this in the text.

8. The rationale for choosing lines for scRNA-seq (Etv2:kaede, Fli1:dsRed; Tp1:GFP, and Draculin:Dendra:H2B) is not clear because these lines are not specific to ECs and are known collectively to mark blood, endocardium, and other cell types. How do the authors distinguish between subtypes of ECs versus other cell types altogether?

We chose a variety of lines to ensure we would mark all EC subsets and have reference populations built into our data sets. For example, *etv2:Kaede* is a general marker of endothelium, *fli1:DsRed/tp1:GFP* marks arterial endothelium, and *draculin* (*drl*) labels LPM, ECs, and HECs depending upon the developmental timepoint. Having distinct overlapping and nonoverlapping populations allowed us to validate our results via the presence or absence of the respective populations of interest. For example, SDECs do not appear in *drl*-marked cells as this line marks LPM descendants. Following unbiased clustering of all ECs, we used the coexpression of landmark genes to assign ECs with different tissue origins (Figure 1 —figure supplement 1 and source data 1). We have clarified this in the text.

9. The authors could show in Figure 6 where etv2:kaede and the other sorted reporter lines map onto these clusters. The authors could also provide scRNA-seq data for the same etv2:gfp+ cells that were presented in previous figures to confirm the presence of SDECs.

Good ideas. We have revised Figure 6 (now Figure 1) to include a UMAP and heatmap with EC cluster identity and differentially expressed genes used to help identify each cluster.

10. A recent paper comparing a new etv2 knock-in line with existing transgenic lines shows differences in expression patterns (Chestnut & Sumanas Dev Dyn 2020). Although data is presented in this study that shows endogenous expression of etv2 using FISH, it is possible that expression and conclusions drawn from the etv2:gfp line does not precisely follow endogenous cell types. This caveat should be considered and discussed within the larger context of the study.

See response to comment 8. Because each EC marker captures only a subset of the total EC populations, we used a broader strategy to capture and analyze ECs, by sorting ECs not only from *TgBAC(etv2:Kaede)^ci6^ embryos* but also from *Tg(fli1:eGFP)^y1^; Tg(tp1:GFP)^um14^, and Tg(drl:H2B-dendra).* For imaging and lineage-based experiments, we have used a validated version of *etv2* reporter line, *Tg(etv2.1:eGFP)^zf372^* (Veldman and Lin, 2012) that, was shown to be more precise than the previous BAC-based *etv2* reporter line. In addition, most of our major conclusions based upon the *Tg(etv2.1:eGFP)^zf372^* reporter lines are complemented by our FISH studies for endogenous *etv2* expression; we observed close correlations with the transgenic pattern in all cases analyzed.

11. It is unclear from the FISH images if the etv2+ cells are in fact double positive with meox1, or if etv2+ cells are intercalated between meox1 cells. The green meox1 signal appears low within the boundaries of etv2+ cells. Could this be resolved by crossing the etv2:gfp line with a red muscle progenitor reporter line, if available?

We do not have a red reporter line for muscle cells available. However, our results show coexpression of *etv2* with *pax3a* (Figure 4F); thus, both experiments support that SDECs are derived from shared progenitors.

12. The authors do not discuss the paper by Murayama et al. (Nat. Comm. 2015) that shows stromal cells in the caudal hematopoietic tissue (CHT) directly delaminate from the somites. There are important similarities between this manuscript and the previous study that are not explored or referenced. For example, in Figure 5G, what are the round cells in the CHT that have switched? Could these be the stromal cells observed by Murayama et al? The switched cells in Figure 5J have endothelial morphology, but not the cells in Figure 5G.

We believe the cells under study in the Murayama paper are vascular smooth muscle cells or stromal cells based upon their morphology and lineal origin from the sclerotome. We thus did not think the papers were directly comparable. That said, on some rare occasions, we occasionally detected lineage-traced cells in the posterior caudal vein, intersomitic vessels or extravascular space. We now discuss these findings and the possibility of having labeled some of the stromal cells identified by Murayama and colleagues.

13. The authors say that lineage tracing of tbx6:cre cells in the DA are consistent between multiple embryos over different stages, however this should be better quantified with more of the data represented in the analysis. It would be informative to show data before the 90 hpf timepoint in Figure 5E-J.

Agreed. We have now included photoconvertible *Tg(actb2:nls-Eos); Tg(fli1:eGFP)^y1^* transgenic animals that describe the migration of the SDECs with greater precision and show their contribution to the DA at 36 hpf (Figure 3). In addition, our imaging experiments using *tbx6:Gal4; Tg(UAS-Cre); A2BD* show PM-derived SDECs incorporated into the floor of the DA at 48 hpf (Figure 8A-C).

14. Are the Cre reporter lines single insertions? It appears from the FACs plots in S3 there are a range of signal intensities, and that some cells may express both colors, suggesting incomplete recombination of multiple insertions. This is possible too in Figure 5G that shows cell that appear double positive. Are there additional controls that could be presented?

We found that all *cre* lines utilized segregated in mendelian ratios, consistent with only single insertions. In addition, we found that the range of signal intensities in the switched RFP^+^ (Figure 8 —figure supplement 1A) is mirrored by the range of intensities observed in the unswitched BFP^+^ cells (Figure 8 —figure supplement 1B). Therefore, we believe that fluorophore expression levels are based on the hematopoietic cell type in which they are expressed and not on transgene copy number.

15. The draculin-cre result is not surprising as the line markers endothelial and blood precursors that, as expected, are labeled in this experiment. These results are independent of SDEC ontogeny. This should be clarified.

The *drl:Cre*^ERT2^ results were added as a control / contrast to those obtained with *tbx6:Cre*. We have clarified this information in the text.

16. In Figure 7A, bmp4 only appears upregulated in a small number of SDECs, and yet the conclusion is made that this is an important factor produced by SDECs.

It is true that *bmp4* is only expressed in a few SDECs. However, we observe no *bmp4* expression in endothelial cells from the LPM and believe that this difference could be due to heterogeneity within SDECs and/or resolution of the scRNA-Seq datasets. That said, we also felt the *bmp4* results should be further investigated and reported in future work. We have removed the results on *bmp4* and *id1* from this manuscript.

[Editors' note: further revisions were suggested prior to acceptance, as described below.]

The manuscript has been improved but there are some remaining issues that need to be addressed, as outlined below:The reviewers are very enthusiastic about the revised manuscript. The only outstanding issue is the presentation of new data that does not have quantification or statistical analysis.Reviewer #3 (Recommendations for the authors):In this study, the authors identify an additional source of dorsal aorta endothelium derived from the somites, which are conventionally thought of as defined blocks of skeletal muscle. These cells in the dermamyotome are positive for endothelial cell markers and are proposed as bipotential precursors that can give rise to endothelium or muscle. The authors conclude that these cells are supportive of HSC emergence, but do not themselves produce HSCs.Most of the previous reviewers' comments have been addressed. However, there are new imaging data presented that has not been analyzed quantitatively or rigorously tested for significant changes using statistical tests.1. Quantification of results in Figure 4F,G, 5A-F, 5I-N, 7C-D.

Figure number 4F and 4G: For each experiment, sections from 6 independent embryos were taken. We observed 1-2 *etv2* positive cells per somite in each of the embryos examined.

Figure 5A-F: For each experiment, sections from 3 independent embryos were taken. We observed 3-4 *etv2* positive cells per somite in the *meox1* morphants compared to 1-2 *etv2* positive cells per somite in the siblings.

Figure 5I-K: For each experiment, sections from 5 independent embryos were taken. We observed 3-4 *etv2* positive cells per somite in the *meox1* morphants and >8 *etv2* positive cells in the *meox1*; Mib double morphants.

Figure 5L-N: For each experiment, sections from 3 independent embryos were taken. We observed 2-4 *etv2* positive cells per somite in the *notch3* mutants and >6 *etv2* positive cells in the *notch3* mutants; Mib morphants.

Figure 7C-D: The number of repeats for the control (n=7) and the WNT inhibitor (n=5) is already depicted in the image. For the WNT inhibited embryos, 4 out of 5 showed abnormal migration of the SDECs.

2. Is decrease in meox1 significant in Figure 7B? If not, this should be clarified.

We thank the reviewer for his comment. We repeated the statistical analysis of the WNT inhibitor versus sibling data set. We observed decreased expression of *axin2* with a concomitant increase in *etv2* by qRT-PCR. In addition, we observed a reduction in *meox1* expression, although not statistically significant. All genes analyzed between Wnt inhibitor and sibling embryos, except *meox1*, showed a statistically significant difference (p<0.001, unpaired, two-tailed Student's t-test; n=3.). We updated the, text, figure legend and methods section.

3. Need more detail about microscopy that allowed resolution of CFP and YFP from kdrl:CSY line in Figure 8E-J.

We thank the reviewer for his comment. We added the following description of the imaging settings and filters used to distinguish between CFP and YFP in the material and methods sections. “Since the emission spectra of CFP and YFP overlap, images were captured in two separate sequences to filter overlapping signals by limiting the PMT detection. For CFP a 476nm laser was used, and the PMT detector was set to collect signals ranging between 480nm and 505nm. For YFP, a 514nm laser was used, and the PMT detector was set to collect signals ranging between 520nm and 570nm.”

4. The drl:Cre-ERT2 data in Figure 8H-K is not clearly convincing and could be removed. The tbx6:Cre data makes the point well.

We thank the reviewer for his comment. Draculin is an early broad marker gene for the lateral plate mesoderm. Therefore using the drl:Cre^ERT2^ as a driver for converting endothelial cells is a good control to show the potential of these cells to convert, hence, the efficiency of the reporter line. Thus, the data in Figure 8 serves not only to show that most cells in the DA are descendants of LPM but also as control for our kdrl:CFP to YFP reporter fish line.